# Scalable Constrained Multi-Agent Reinforcement Learning via State Augmentation and Consensus for Separable Dynamics

**Santiago Amaya-Corredor**                                        *santiagoesteban.amaya@upf.edu*
*Department of Engineering*
*University Pompeu Fabra*

**Miguel Calvo-Fullana**                                           *miguel.calvo@upf.edu*
*Department of Engineering*
*University Pompeu Fabra*

**Anders Jonsson**                                                 *anders.jonsson@upf.edu*
*Department of Engineering*
*University Pompeu Fabra*

**Reviewed on OpenReview:** *https://openreview.net/forum?id=whihxstZcO*

## Abstract

We present a distributed approach for constrained Multi-Agent Reinforcement Learning (MARL) that combines state-augmented policy learning with distributed consensus over dual variables. Our method targets systems where agents have separable dynamics but must coordinate to satisfy global resource constraints, a setting in which, as we demonstrate empirically, independent learning fails to produce feasible solutions because agents cannot determine appropriate individual contributions toward collective constraint satisfaction. The key technical contribution is showing that lightweight neighbor-to-neighbor consensus over Lagrange multipliers suffices for globally coordinated constraint enforcement while preserving the scalability of independent training. Each agent learns a single augmented policy offline, conditioned on both its local state and a dual variable encoding constraint feedback. During execution, agents reach agreement on this dual variable through local communication alone. We prove that under mild connectivity assumptions, the consensus error among agents' multipliers is bounded, and show that this translates to a bounded constraint violation that decreases with graph connectivity and the number of consensus rounds. Unlike centralized training with decentralized execution (CTDE) approaches, whose complexity grows at least quadratically with agent count, our method scales linearly in both training and execution. Experiments on smart grid demand response demonstrate that consensus coordination is *essential for feasibility*: without it, agents satisfy grid capacity constraints only by indefinitely postponing demand, a degenerate non-solution. With consensus, agents converge to a shared dual variable and satisfy both grid constraints and demand fulfillment, scaling to thousands of agents while CTDE baselines are limited to dozens.

## 1 Introduction

In recent years, reinforcement learning (RL) has achieved significant success in solving diverse and complex decision-making tasks (Brown & Sandholm, 2019; Orr & Dutta, 2023; Silver et al., 2017). Many of these successes involve multiple agents and can be characterized as multi-agent RL (MARL). Generally, MARL addresses a sequential problem where a set of autonomous agents make decisions and interact in a shared environment to maximize a reward. However, MARL problems can quickly become intractable as the number

of agents increases, since the number of possible interactions and the space of possible states can grow exponentially in the number of agents. Moreover, as all agents navigate and learn simultaneously, the environment may become non-stationary, invalidating many of the single-agent RL assumptions. In realistic scenarios, conflicting objectives often need to be balanced to achieve satisfactory solutions. This issue is exacerbated when increasing the number of autonomous agents, whose specific goals are not commonly aligned. Finding optimal strategies in multi-agent systems (MAS) usually require at least some level of coordination and communication.

Our work addresses distributed systems where agents have separable dynamics but must coordinate to satisfy global operational constraints. While this assumption is restrictive compared to general MARL with coupled dynamics, it enables linear scaling in both training and execution, making our approach practical for hundreds or thousands of agents. The setting remains genuinely multi-agent as agents must coordinate through consensus to satisfy global constraints. This structure naturally arises in infrastructure management (e.g., building thermostats, EV chargers) where local controllers make independent decisions but must respect system-wide limits (e.g., power grid capacity). When agents share the same MDP, we only need to train one policy for all agents, significantly reducing complexity. The multi-agent coordination occurs through consensus on a dual variable during execution. Our Constrained MARL (CMARL) framework has each agent maximize a primary reward while adhering to a global average constraint on a secondary reward, with the constraint acting as the coupling mechanism.

Agents communicate only with immediate neighbors in the network, reflecting realistic constraints where global broadcast is infeasible. Through local communication, agents share dual variables to achieve consensus dynamically. Crucially, we demonstrate empirically that this coordination is not merely beneficial but *necessary*: without consensus, independently-trained agents cannot distinguish between feasible demand management and trivial constraint satisfaction through indefinite demand postponement. The dual variable consensus mechanism is what transforms independently-trained policies into a collectively feasible solution. We develop a novel CMARL algorithm and validate it on smart grid management (Dileep, 2020), optimizing energy distribution while satisfying operational constraints. Our experiments demonstrate scalability across different network configurations with varying complexity and agent heterogeneity. The contributions of this paper are threefold:

1. **A distributed consensus mechanism for constrained MARL with separable dynamics.** We integrate distributed consensus over Lagrange multipliers with state-augmented constrained RL. We prove bounded consensus error under mild connectivity assumptions and show that this implies bounded constraint violation through a Lipschitz sensitivity argument. This enables global constraint coordination through local communication alone, without centralized critics or joint state observations.

2. **Linear scalability in both training and execution.** By exploiting problem separability for policy learning and lightweight consensus for coordination, our method scales to 1000 agents while CTDE approaches are limited to tens of agents due to centralized critic requirements.

3. **Empirical evidence that consensus coordination is necessary for feasibility.** On smart grid economic dispatch, we show that the same independently-trained policies produce fundamentally different outcomes, feasible versus infeasible, depending solely on whether consensus is enabled. We further validate the approach against a centralized oracle, demonstrating that distributed consensus achieves equivalent constraint satisfaction and cost ($<0.2\%$ gap).

## 2 Related Work

**Constrained Reinforcement Learning.** Constrained RL formulates objectives with side constraints, typically solved via Lagrangian methods (Altman, 2021; Borkar, 2005). We distinguish two paradigms: *safe RL* ensures per-step constraint satisfaction (Achiam et al., 2017; Chow et al., 2019; Achiam & Amodei, 2019), while *average-constrained RL* permits temporary violations if long-term averages stay within bounds (Liang et al., 2018; Paternain et al., 2022). The latter suits resource management where momentary spikes are

acceptable. Standard Lagrangian methods can fail to produce feasible policies when the relationship between dual variables and constraint satisfaction is complex. State augmentation (Calvo-Fullana et al., 2024) addresses this by incorporating the dual variable into the state representation, enabling policies that adapt their behavior to the current level of constraint pressure. Our work extends state augmentation from the single-agent to the multi-agent setting through distributed consensus on the shared dual variable; concurrent multi-agent state-augmentation work (Agorio et al., 2024) addresses assignment problems.

**Cooperative MARL.** Centralized training with decentralized execution (CTDE) has emerged as the dominant paradigm for cooperative MARL (Kraemer & Banerjee, 2016), encompassing value decomposition methods (Sunehag et al., 2018; Rashid et al., 2018; Wang et al., 2021) and policy gradient approaches with centralized critics (Lowe et al., 2017; Yu et al., 2022), though both scale poorly with agent count. Recent work has questioned whether CTDE provides benefits over independent learning in many settings (de Witt et al., 2020), though independent methods cannot handle global constraints without additional mechanisms. For comprehensive coverage, we refer to recent surveys (Cheruiyot et al., 2025; Low & Zhou, 2025; Hady et al., 2025) and the textbook by Albrecht et al. (2024). Our approach differs fundamentally: rather than centralizing during training, we train policies independently and coordinate only through dual variable consensus during execution.

**Networked and Scalable MARL.** Several works achieve scalability by exploiting network structure. Zhang et al. (2018) develop fully decentralized MARL for networked agents, establishing convergence for policy gradient methods with local communication. Qu et al. (2020) propose scalable multi-agent RL for networked systems with average reward objectives, achieving linear scaling through separable dynamics but without handling constraints. Chu et al. (2020) develop a multi-agent RL framework for networked system control with localized rewards. In the power systems domain, Chen et al. (2022) develop PowerNet for scalable grid control, Feng et al. (2024) address stability-constrained RL for decentralized voltage control, and Mai et al. (2024) study multi-agent RL for fast-timescale demand response. More recently, Wang et al. (2025) propose distributed MARL for multi-objective microgrid dispatch using actor-critic with multiple critics, achieving coordination through neighbor-to-neighbor communication; however, their method targets Pareto optimization without explicit global constraint handling. While these works demonstrate scalability through network structure, none provides mechanisms for satisfying global constraints that couple agents' decisions.

**Constrained Multi-Agent RL.** Recent work addresses constraints in MARL with coupled dynamics. Lu et al. (2021) propose Safe Dec-PG for distributed constrained MDPs, achieving convergence but with computational complexity limiting experiments to 5 agents. Gu et al. (2021) introduce Multi-Agent Constrained Policy Optimization (MACPO), extending TRPO-Lagrangian to multi-agent settings. Ying et al. (2023) develop a primal-dual actor-critic method using $\kappa$-hop truncation, trading off constraint coupling accuracy for scalability. Zhang et al. (2024) introduce Scal-MAPPO-L with local policy optimization, though they note exponential growth in state-action space with neighborhood size (experiments limited to 12 agents). On the theoretical front, Chen et al. (2024) establish hardness results showing constrained cooperative MARL is computationally intractable in general settings, while Ding et al. (2023) provide a generalized Lagrangian primal-dual scheme with provable sample-efficient convergence. Importantly, these methods target *safety constraints* requiring per-step satisfaction under *coupled dynamics* where agents' transitions directly affect each other. While more general than our setting, this generality comes at computational cost that limits scalability to tens of agents. We target *average constraint satisfaction* with *separable dynamics*—assumptions satisfied in many infrastructure management problems—enabling linear scaling demonstrated up to 1000 agents.

**Distributed Optimization and Consensus in RL.** Our dual consensus mechanism builds on distributed optimization with coupled constraints (Nedic & Ozdaglar, 2009; Olfati-Saber et al., 2007), with recent extensions to RL settings (Yarmoshik et al., 2024; Wang et al., 2024). Closely related, Oh et al. (2025) develop consensus-based actor-critic for network optimization, where agents share *local rewards* with neighbors to approximate the global return without centralized training. Our consensus operates on a fundamentally different quantity: *dual variables* (Lagrange multipliers) rather than rewards. In reward-consensus methods (Oh et al., 2025; Wang et al., 2025), each agent's feasibility is independent, whereas our global constraint $\sum_i V_1^i(\pi^i) \leq c$ couples agents' feasible sets. This requires state-augmented policies $\pi^i(s, \lambda)$ that respond to

Table 1: Comparison of constrained and scalable MARL methods. Scalability indicates computational complexity: exponential in agents ($n$), exponential in communication radius ($\kappa$), or linear. Methods marked with † handle coupled dynamics but at higher computational cost.

| Method | Constraints | Dynamics | Scalability | Max Agents |
|---|---|---|---|---|
| *Scalable (separable/networked dynamics, no global constraints)* | | | | |
| Qu et al. (2020) | None | Networked | Linear | 25 |
| Chu et al. (2020) | None | Networked | Linear | 28 |
| Chen et al. (2022) | None | Networked | Linear | 40 |
| Feng et al. (2024) | Stability (local) | Networked | Linear | 123 |
| *Constrained (coupled dynamics, limited scalability)*† | | | | |
| Lu et al. (2021) | Safety (per-step) | Coupled | Exp. in $n$ | 5 |
| Gu et al. (2021) | Safety (per-step) | Coupled | Exp. in $n$ | 8 |
| Ying et al. (2023) | Safety (per-step) | Coupled | Exp. in $\kappa$ | 20 |
| Zhang et al. (2024) | Safety (per-step) | Coupled | Exp. in $\kappa$ | 12 |
| **Ours** | Average (global) | Separable | Linear | **1000** |

the evolving multiplier—a design absent in reward-consensus methods. As we demonstrate in Section 7.2, this coordination is necessary for feasibility.

**Neural Controllers for Power Systems.** Cui et al. (2023) propose structured neural-PI controllers using strictly convex neural networks, achieving stability and output tracking through equilibrium-independent passivity—a model-based approach requiring knowledge of system dynamics. Feng et al. (2023) construct decentralized RL-based controllers for voltage regulation, combining a transient neural policy with a steady-state gradient flow optimizer. While both address power systems with distributed controllers, they focus on *stability and tracking* rather than *constraint satisfaction*, embedding constraints in the controller architecture rather than handling them explicitly through Lagrangian duality. The approaches are complementary.

**Positioning Our Work.** Table 1 summarizes how our approach compares to prior methods across three dimensions: constraint type, dynamics structure, and scalability. Our key distinction is the combination of (i) separable dynamics enabling independent policy training, (ii) average rather than per-step constraint satisfaction, and (iii) distributed consensus over dual variables for coordination. This combination enables linear scaling—demonstrated up to 1000 agents—while methods handling coupled dynamics (Lu et al., 2021; Ying et al., 2023; Zhang et al., 2024) are limited to tens of agents, and methods achieving scalability through separable structure (Qu et al., 2020; Chu et al., 2020) lack explicit constraint handling. For the important class of infrastructure management problems satisfying our assumptions, we achieve unprecedented scale with bounded constraint violation guarantees, coordinated through lightweight neighbor-to-neighbor communication.

## 3 Problem Formulation

Typically, CMARL is studied using the Markov Games framework (Littman, 1994), an extension of game theory to environments where the dynamics can be modeled using a Markov Decision Process (MDP). Markov games model interactions among multiple agents whose decisions influence a shared environment. In our distributed constrained setting, the Markov game is defined by the tuple $\langle N, \{S^i\}_{i=1}^N, \{A^i\}_{i=1}^N, \{P^i\}_{i=1}^N, \{r_0^i\}_{i=1}^N, \{r_1^i\}_{i=1}^N \rangle$, where $N$ is the number of agents, $S^i \subset \mathbb{R}^m$ and $A^i \subset \mathbb{R}^d$ are compact sets denoting the states and actions of agent $i$, with $S := S^1 \times \cdots \times S^N$ and $A := A^1 \times \cdots \times A^N$ denoting the sets of joint states and actions. The joint state transition probability is given by $P : S \times A \to \Delta(S)$, with each individual agent's transition given by $P^i : S^i \times A^i \to \Delta(S^i)$, where $\Delta(S)$ is the probability simplex on $S$. We further denote by $r_0^i : S^i \times A^i \to \mathbb{R}$ the reward function for the main objective of agent $i$, and by $r_1^i : S^i \times A^i \to \mathbb{R}$ the reward function for the secondary objective subject to a constraint, with global counterparts defined as $r_0 : S \times A \to \mathbb{R}$ and $r_1 : S \times A \to \mathbb{R}$. At time $t$, given a joint state $s_t = (s_t^1, \ldots, s_t^N)$ and action $a_t = (a_t^1, \ldots, a_t^N)$, the system transitions to a new state $s_{t+1} = (s_{t+1}^1, \ldots, s_{t+1}^N)$ with probability $P(s_{t+1}|s_t, a_t)$. The Markov property ensures that the system

dynamics only depend on the last state and action, i.e. $P(s_{t+1}|s_0, a_0, \ldots, s_t, a_t) = P(s_{t+1}|s_t, a_t)$. We consider a scenario where agents maximize a primary reward $r_0$ subject to a global resource constraint on a secondary quantity $r_1$. Specifically, we aim to maximize the long-term average of $r_0(s_t, a_t)$, while ensuring that the long-term average of $r_1(s_t, a_t)$ does not exceed a given upper bound $c$.[1] This constrained optimization problem can be expressed as

$$\underset{\pi}{\text{maximize}} \quad \lim_{T \to \infty} \frac{1}{T} \mathbb{E}_{s,a \sim \pi} \left[ \sum_{t=0}^{T} r_0(s_t, a_t) \right] \tag{1a}$$

$$\text{subject to} \quad \lim_{T \to \infty} \frac{1}{T} \mathbb{E}_{s,a \sim \pi} \left[ \sum_{t=0}^{T} r_1(s_t, a_t) \right] \leq c. \tag{1b}$$

This is a multi-agent centralized problem, which is often impractical due to its poor scalability. Specifically, we are interested in problems that can be decomposed into distributed problems. Formally, we consider scenarios satisfying the following assumptions.

**Assumption 3.1** (Independent Policies). Each agent $i$ selects an action taking into account only its own local state. Namely, $\pi(a_t|s_t) = \prod_{n=1}^{N} \pi^n(a_t^n|s_t^n)$.

**Assumption 3.2** (Separable Dynamics). The actions of one agent do not affect the states of others. That is, state transitions are given by $P(s_{t+1}|s_t, a_t) = \prod_{i=1}^{N} P^i(s_{t+1}^i|s_t^i, a_t^i)$.

**Assumption 3.3** (Summable Rewards). The global reward can be decomposed as the sum of individual rewards, i.e. $r_0(s_t, a_t) = \sum_{n=1}^{N} r_0^n(s_t^i, a_t^i)$ and $r_1(s_t, a_t) = \sum_{n=1}^{N} r_1^n(s_t^i, a_t^i)$.

*Remark* 3.4 (Scope and Limitations). Assumptions 3.1-3.3 significantly restrict the class of problems we address. Under these assumptions, agents do not influence each other's states or rewards directly, which excludes many classical MARL scenarios like multi-robot coordination or competitive games. However, these assumptions are satisfied in important real-world domains:

- **Smart Grid Management**: Buildings independently control their energy consumption but share grid capacity constraints.

- **Distributed Computing**: Processes independently execute but share memory/bandwidth limits.

- **Fleet Management**: Vehicles independently plan routes but share depot capacity or charging infrastructure constraints.

For problems where agents' dynamics are coupled—such as traffic congestion, where one vehicle's actions affect others' travel times—methods like those of Lu et al. (2021) and Zhang et al. (2024) are more appropriate, albeit at higher computational cost. Our contribution is demonstrating that when separable structure holds, it can be exploited for orders-of-magnitude improvements in scalability.

The first assumption allows each agent to operate based solely on local information, the second assumption ensures that the interactions of the agents are structured in a non-interfering manner, and the third assumption ensures that global objectives can be achieved through local decisions. This set of assumptions allows for the problem to be rewritten in the following form:

$$\max_{\pi^1, \ldots, \pi^N} \sum_{i=1}^{N} \lim_{T \to \infty} \frac{1}{T} \mathbb{E}_{s^i, a^i \sim \pi^i} \left[ \sum_{t=0}^{T} r_0^i(s_t^i, a_t^i) \right] \tag{2a}$$

$$\text{s.t.} \sum_{i=1}^{N} \lim_{T \to \infty} \frac{1}{T} \mathbb{E}_{s^i, a^i \sim \pi^i} \left[ \sum_{t=0}^{T} r_1^i(s_t^i, a_t^i) \right] \leq c. \tag{2b}$$

By defining value functions as the long-term average of each reward,

$$V_j^i(\pi^i) \triangleq \lim_{T \to \infty} \frac{1}{T} \mathbb{E}_{s^i, a^i \sim \pi^i} \left[ \sum_{t=0}^{T} r_j^i(s_t^i, a_t^i) \right], \tag{3}$$

---

[1]For simplicity, we restrict ourselves to the single constraint case, though the results generalize to multiple constraints.

we can then rewrite the maximization problem in equation 2 in the following more concise manner:

$$\underset{\pi^1,\ldots,\pi^N}{\text{maximize}} \sum_{i=1}^{N} V_0^i(\pi^i) \text{ subject to } \sum_{i=1}^{N} V_1^i(\pi^i) \leq c. \tag{4}$$

The resulting formulation now exhibits a certain degree of separability across agents, with each agent maximizing its own policy with respect to its individual value function, while still being coupled to the other agents through the global constraint. While the separable structure might suggest independent single-agent solutions would suffice, the global constraint in equation 4 creates a critical coordination challenge: without communication, agents cannot determine appropriate individual contributions to satisfy the collective constraint. This necessitates our consensus mechanism to coordinate the dual variables that encode constraint violation feedback.

## 4 Methodology

We begin by formulating the Lagrangian of the optimization problem in equation 4. This involves introducing Lagrange multipliers to transform the constrained optimization problem into a form where the constraints are incorporated into the objective function as penalty terms. Namely,

$$\mathcal{L}(\pi, \lambda) = \sum_{i=1}^{N} V_0^i(\pi^i) + \lambda\left(c - \sum_{i=1}^{N} V_1^i(\pi^i)\right) = \sum_{i=1}^{N}\left(V_0^i(\pi^i) + \lambda\left(\frac{c}{N} - V_1^i(\pi^i)\right)\right), \tag{5}$$

where $\lambda \in \mathbb{R}^+$ is the Lagrange multiplier (dual variable) for the inequality constraint. We rewrite the Lagrangian as individual agent components to maintain distributed formulation. The dual problem becomes

$$\underset{\lambda}{\text{minimize}} \left[\sum_{i=1}^{N} \max_{\pi^i}\left(V_0^i(\pi^i) + \lambda\left(\frac{c}{N} - V_1^i(\pi^i)\right)\right)\right] \tag{6}$$

where summation and maximization are exchanged due to Assumptions 3.1 and 3.2. This decomposition enables independent local optimization while satisfying the global constraint. The problem exhibits strong duality (Paternain et al., 2019), so the optimal solution of equation 4 equals the saddle-point of equation 6.

*Remark* 4.1 (Comparison with Standard Primal-Dual Methods). Standard distributed primal-dual methods (Yarmoshik et al., 2024; Wang et al., 2024) require strongly convex objectives or extensive message passing. Our approach differs by: (i) using state augmentation from single-agent CRL (Calvo-Fullana et al., 2024) for non-convex policy optimization, and (ii) requiring only single-scalar neighbor communication. Integrating standard consensus (Xiao & Boyd, 2003) with state-augmented RL policies enables our scalability.

### 4.1 Offline independent training

The primal step of equation 6 (policy learning) is distributable. For a given $\lambda$, the problem decomposes across agents. Defining the weighted reward $r_\lambda^i(s_t^i, a_t^i) \triangleq r_0^i(s_t^i, a_t^i) - \lambda r_1^i(s_t^i, a_t^i)$, the maximization becomes

$$\{\pi_\star^i(\lambda)\} = \underset{\pi^1,\ldots,\pi^N}{\arg\max} \sum_{i=1}^{N} \lim_{T\to\infty} \frac{1}{T} \mathbb{E}_{s^i, a^i \sim \pi^i}\left[\sum_{t=0}^{T} r_\lambda^i(s_t^i, a_t^i)\right]. \tag{7}$$

Each agent's primal step follows standard unconstrained RL. However, standard dual methods can fail to produce feasible policies for CRL (Calvo-Fullana et al., 2024). We thus learn state-augmented policies $\pi^i(\lambda)$ in augmented space $S^i \times \mathbb{R}_+$ that maximize equation 5, instead of ordinary policies in $S^i$. Agents independently train these policies using any standard RL method. The trained policies handle any constraint level $c$ when coupled with our dual update mechanism.

### 4.2 Online dual consensus

Determining $\lambda$ remains challenging since gradient descent on equation 5 couples all agents. Consider agents communicating over an undirected graph $G = (V, E)$, where $V$ are vertices (agents) and $E \subset N \times N$ are edges. The neighborhood $\mathcal{N}^i = \{j \in V \mid (i, j) \in E\}$ contains nodes directly connected to $i$. We rewrite equation 6 in distributed dual consensus form:

$$\operatorname*{minimize}_{\lambda^1, \ldots, \lambda^N} \sum_{i=1}^{N} \max_{\pi^i} \left( V_0^i(\pi^i) + \lambda^i \left( \frac{c}{N} - V_1^i(\pi^i) \right) \right) \tag{8a}$$

$$\text{subject to} \quad \lambda^i = \frac{1}{|\mathcal{N}_i|} \sum_{n \in \mathcal{N}^i} \lambda^n, \quad i = 1, \ldots, N. \tag{8b}$$

The solution to equation 8 equals that of equation 6. Each agent holds local copy $\lambda^i$, with neighborhood constraints ensuring consensus. Using optimal policies from equation 7, we obtain

$$\operatorname*{minimize}_{\lambda^1, \ldots, \lambda^N} \sum_{i=1}^{N} \left[ V_0^i\big(\pi_\star^i(\lambda^i)\big) + \lambda^i \left( \frac{c}{N} - V_1^i\big(\pi_\star^i(\lambda^i)\big) \right) \right] \tag{9a}$$

$$\text{subject to} \quad \lambda^i = \frac{1}{|\mathcal{N}_i|} \sum_{n \in \mathcal{N}^i} \lambda^n, \quad i = 1, \ldots, N. \tag{9b}$$

### 4.3 Primal-consensus update

To solve equation 9, each agent $i$ maintains $\lambda^i$ and iteratively (i) performs local gradient updates for constraint satisfaction and (ii) averages with neighbors' variables. With gradient step size $\alpha > 0$ and consensus step size $\epsilon > 0$, agent $i$ updates:

$$\lambda_{k+1}^i = \lambda_k^i - \alpha \nabla_{\lambda^i}\left[ V_0^i\big(\pi_\star^i(\lambda_k^i)\big) + \lambda_k^i \left( \frac{c}{N} - V_1^i\big(\pi_\star^i(\lambda_k^i)\big) \right) \right] - \epsilon \left( \lambda_k^i - \overline{\lambda}_k^i \right), \tag{10}$$

where $\overline{\lambda}_k^i = \sum_{n \in \mathcal{N}_i} \lambda_k^n / |\mathcal{N}_i|$ is the neighbor average. The first term performs local gradient descent; the second enforces consensus. These corrections drive all $\lambda^i$ to converge, matching the solution of equation 6.

*Remark* 4.2 (Relation to Centralized Dual). As agents optimize local $\lambda^i$ while enforcing neighbor consensus, $\{\lambda^i\}$ converges to the same value as the global $\lambda$ in equation 6. Thus, equation 10 provides a fully distributed solution without centralized coordination.

## 5 Algorithm

Each agent $i$ optimizes its local policy by maximizing the Lagrangian given its current copy of the dual variable $\lambda^i$. To ensure that the policy appropriately accounts for constraint satisfaction, we augment each agent's state space with the local multiplier $\lambda^i$. This augmentation yields a policy $\pi_\star^i(s_t^i, \lambda_t^i)$ that views $\lambda^i$ as part of the state, so that standard reinforcement-learning (RL) algorithms can be used to learn this policy.[2]

If we have an optimal policy for a given set of multipliers $\pi_\star(s, \lambda)$, and we *continuously update* these multipliers (via 10), then the state-action trajectories generated by each agent satisfy the constraints in equation 2 (Calvo-Fullana et al., 2024, Theorem 1). Combining these ideas, we summarize the execution in Algorithm 1.

**Theorem 5.1.** *Suppose the local value functions satisfy*

$$\left\| V_1^i\big(\pi_\star^i(\lambda_k^i)\big) - \frac{1}{N} \sum_{j=1}^{N} V_1^j\big(\pi_\star^j(\lambda_k^j)\big) \right\| \leq \sigma, \tag{11}$$

---

[2]In practice, many RL methods—e.g., policy gradient, value-based methods—can be adapted to handle such an augmented state.

and let $w^i = |\mathcal{N}^i| / \sum_{j=1}^N |\mathcal{N}^j|$. Under mild conditions on the connectivity and step sizes, the execution of Algorithm 1 results in a bounded consensus error:

$$\lim_{k \to \infty} \left\| \lambda_{k+1} - \sum_{i=1}^N w^i \lambda_k^i \right\| \leq \frac{\rho^{\mathscr{L}}}{1 - \rho^{\mathscr{L}}} \, \alpha \, \sigma, \tag{12}$$

where $\rho$ and $\mathscr{L}$ relate to the graph's spectral properties and the number of communication steps per iteration (or partial consensus steps).

Theorem 5.1 guarantees that agents' local multipliers remain close to one another throughout execution. The proof appears in Appendix A.1. The bound decreases as $\mathscr{L}$ increases, indicating that additional consensus steps reduce the discrepancy among agents' multipliers. In practice, a small $\rho$ (which occurs in well-connected graphs) accelerates convergence, allowing $\mathscr{L}$ to remain small. For many real-world network structures, a single consensus iteration ($\mathscr{L} = 1$) per gradient step suffices to keep the discrepancy in $\lambda^i$ below an acceptable threshold.

A natural question is whether bounded multiplier disagreement translates into bounded constraint violation and near-optimal performance. We answer this affirmatively under two additional assumptions.

**Assumption 5.2** (Lipschitz Sensitivity of Constraint Value). For each agent $i$, the map $\lambda \mapsto V_1^i\big(\pi_\star^i(\lambda)\big)$ is $L_V$-Lipschitz on $\mathbb{R}_+$:

$$\left| V_1^i\big(\pi_\star^i(\lambda)\big) - V_1^i\big(\pi_\star^i(\lambda')\big) \right| \leq L_V \, |\lambda - \lambda'|, \qquad \forall \, \lambda, \lambda' \geq 0, \quad \forall \, i. \tag{13}$$

**Assumption 5.3** (Approximate Policy Optimality). Each agent's learned policy $\hat{\pi}^i(\lambda)$ is $\varepsilon_{\mathrm{approx}}$-optimal in both value functions: for all $\lambda \geq 0$,

$$\left| V_j^i\big(\hat{\pi}^i(\lambda)\big) - V_j^i\big(\pi_\star^i(\lambda)\big) \right| \leq \varepsilon_{\mathrm{approx}}, \qquad j \in \{0, 1\}, \quad \forall \, i. \tag{14}$$

Assumption 5.2 captures how sensitively each agent's constraint-relevant behavior changes with the dual variable. It is naturally satisfied by state-augmented neural network policies, where $\lambda$ enters as a continuous input to a smooth function approximator; we discuss this in detail in Appendix B.2. Assumption 5.3 accounts for function approximation error due to finite network capacity and finite training.

**Proposition 5.4** (Primal Sensitivity to Consensus Error). Let $\boldsymbol{\lambda} = (\lambda^1, \ldots, \lambda^N)$ be the vector of local dual variables, let $\bar{\lambda} = \sum_{i=1}^N w^i \lambda^i$ denote the degree-weighted average ($w^i = |\mathcal{N}^i| / \sum_j |\mathcal{N}^j|$), and let $\lambda^\star$ denote the centralized dual optimum. Suppose Assumptions 5.2 and 5.3 hold, and the consensus error satisfies $\|\boldsymbol{\lambda} - \bar{\lambda} \mathbf{1}\| \leq \delta$ as guaranteed by Theorem 5.1.

**(a) Feasibility gap due to multiplier disagreement.** The aggregate constraint value under heterogeneous multipliers deviates from that under the common average multiplier by at most

$$\left| \sum_{i=1}^N V_1^i\big(\hat{\pi}^i(\lambda^i)\big) - \sum_{i=1}^N V_1^i\big(\pi_\star^i(\bar{\lambda})\big) \right| \leq L_V \sqrt{N} \, \delta + N \, \varepsilon_{\mathrm{approx}}. \tag{15}$$

**(b) Constraint violation bound.** If the centralized dual optimum yields a feasible solution, i.e. $\sum_{i=1}^N V_1^i\big(\pi_\star^i(\lambda^\star)\big) \leq c$, then the distributed solution satisfies

$$\sum_{i=1}^N V_1^i\big(\hat{\pi}^i(\lambda^i)\big) \leq c + L_V\big(\sqrt{N} \, \delta + N \, |\bar{\lambda} - \lambda^\star|\big) + N \, \varepsilon_{\mathrm{approx}}. \tag{16}$$

**(c) Optimality gap.** If $V_0^i\big(\pi_\star^i(\cdot)\big)$ is also $L_0$-Lipschitz on $\mathbb{R}_+$, then

$$\left| \sum_{i=1}^N V_0^i\big(\hat{\pi}^i(\lambda^i)\big) - \sum_{i=1}^N V_0^i\big(\pi_\star^i(\lambda^\star)\big) \right| \leq L_0\big(\sqrt{N} \, \delta + N \, |\bar{\lambda} - \lambda^\star|\big) + N \, \varepsilon_{\mathrm{approx}}. \tag{17}$$

The proof (Appendix B) combines Lipschitz sensitivity with Cauchy–Schwarz to translate the $\ell_2$ consensus error into an $\ell_1$ bound on individual deviations.

**Corollary 5.5** (Explicit Constraint Violation Bound). *Under the conditions of Proposition 5.4 and Theorem 5.1, if $\bar{\lambda}_k \to \lambda^\star$ and the centralized solution is feasible, then asymptotically the constraint violation is bounded by*

$$\sum_{i=1}^{N} V_1^i\big(\hat{\pi}^i(\lambda_k^i)\big) \;\leq\; c \;+\; \frac{L_V \sqrt{N} \, \rho^{\mathscr{L}} \, \alpha \, \sigma}{1 - \rho^{\mathscr{L}}} \;+\; N \, \varepsilon_{\text{approx}}. \tag{18}$$

*In particular, the constraint violation vanishes as (i) $\mathscr{L} \to \infty$ (more consensus rounds), (ii) $\alpha \to 0$ (smaller dual step size), (iii) $\rho \to 0$ (better-connected graph), or (iv) $\varepsilon_{\text{approx}} \to 0$ (more expressive policy class).*

*Remark* 5.6 (Dual Convergence). The condition $\bar{\lambda}_k \to \lambda^\star$ in Corollary 5.5 holds under standard diminishing step-size schedules for projected subgradient methods on the convex dual problem (Nedic & Ozdaglar, 2009). In our experiments we use a fixed step size, so $|\bar{\lambda} - \lambda^\star|$ remains bounded but non-vanishing; however, Figure 8b shows empirically that the converged multiplier value closely approximates the centralized optimum.

We note that Danskin's theorem also justifies the multiplier update in Algorithm 1. For every fixed multiplier $\lambda$, the policy $\pi_\star(\lambda)$ is *defined* as a maximizer of the inner problem $\max_\pi \mathcal{L}(\pi, \lambda)$. By Danskin's theorem, the gradient of this maximized objective with respect to $\lambda$ depends only on the partial derivative of $\mathcal{L}$, evaluated at the maximizer. Hence, in the multiplier update (Lines 6–8 of Algorithm 1) we treat $\pi_\star$ as constant without loss of correctness. This argument is standard in Lagrangian-based constrained RL (see, e.g., Calvo-Fullana et al., 2024).

---

**Algorithm 1** Distributed multiplier update with Separated Consensus and Gradient Steps

---

1: **Input:** Trained policies $\pi_\star^i(\lambda)$, learning rates $\alpha$, $\epsilon$, requirement $c$, number of consensus steps $\mathscr{L}$
2: **Output:** Trajectories satisfying the constraints
3: **Initialize:** Dual variables $\lambda_0^i = 0$, $\mu_0^i = 0$ for $i = 1, \dots, N$
4: **for** $k = 0, 1, \dots, K-1$ **do**
5:     **Gradient Descent Step:**
6:     $\lambda_{k+\frac{1}{2}}^i = \left[ \lambda_k^i - \alpha \left( \frac{c}{N} - V_{1,k}^i \right) \right]_+$
7:     **Initialize Consensus Variable:**
8:     $\lambda_{\ell=0}^i = \lambda_{k+\frac{1}{2}}^i$
9:     **for** $\ell = 0, \dots, \mathscr{L}-1$ **do**
10:         **Consensus Update:**
11:         $\lambda_{\ell+1}^i = \lambda_\ell^i - \epsilon \left( \lambda_\ell^i - \frac{1}{|\mathcal{N}_i|} \sum_{j \in \mathcal{N}_i} \lambda_\ell^j \right)$
12:     **end for**
13:     **Update for Next Iteration:**
14:     $\lambda_{k+1}^i = \lambda_{\ell=\mathscr{L}}^i$
15: **end for**

---

In practice, we perform only one consensus iteration per time step ($\mathscr{L} = 1$).

*Remark* 5.7 (Instantiation of $V_{1,k}^i$). In Algorithm 1, $V_{1,k}^i = r_1^i(s_k^i, a_k^i)$ is the constraint reward observed by agent $i$ at timestep $k$, serving as a single-sample stochastic estimate of $V_1^i(\pi_\star^i(\lambda_k^i))$. This corresponds to performing stochastic (sub)gradient descent on the dual (Nedic & Ozdaglar, 2009), where the consensus mechanism ensures that the resulting noisy updates remain coordinated across agents.

# 6 Use Case: Smart Grid Management

We apply our method to Demand Response (DR) in a district of buildings with solar energy and battery storage. Our goal is to minimize energy costs for each building while avoiding critical grid peaks through energy storage and load shifting. Each building's agent observes the current demand, battery charge, and grid price, then decides how to allocate energy between grid and battery sources. The local objective is defined as $r_0^i(s_t^i, a_t^i) = -e_{\text{grid}}(s_t^i, a_t^i)\, p_t$, where $e_{\text{grid}}(s_t^i, a_t^i)$ is the building's grid consumption and $p_t$ is the

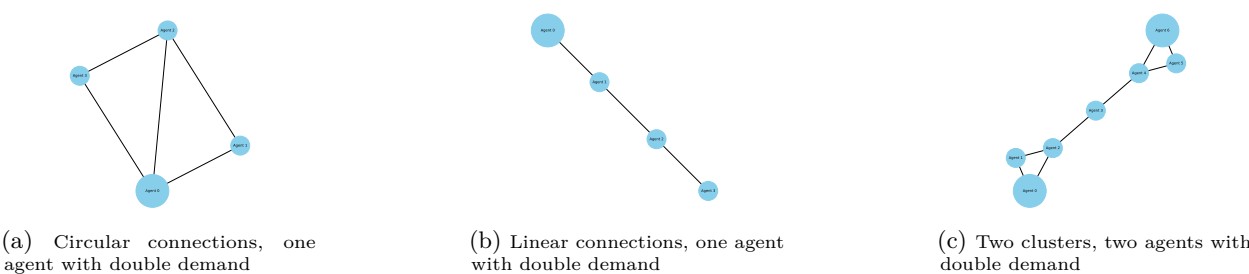

(a) Circular connections, one agent with double demand

(b) Linear connections, one agent with double demand

(c) Two clusters, two agents with double demand

Figure 1: Different communication networks and agent demands.

energy price at time $t$. By maximizing $r_0^i$, agents minimize their grid electricity spending while respecting global consumption constraints.

The secondary reward $r_1^i(s_t^i, a_t^i) = e_{\text{grid}}(s_t^i, a_t^i)$ with constraint $\sum_{i=1}^N V_1^i(\pi^i) \le c$ ensures that average total grid usage stays below threshold $c$ (a percentage of peak demand), maintaining grid stability. Agents can postpone unmet demand for later, and batteries charge automatically from solar generation. To ensure all demand is eventually met, we add a local constraint with reward

$$r_2^i(s_t^i, a_t^i) = d_t^i - e_{\text{grid}}(s_t^i, a_t^i) - e_{\text{bat}}(s_t^i, a_t^i), \tag{19}$$

where $d_t^i$ is the demand of agent $i$ at time $t$ and $e_{\text{bat}}(s_t^i, a_t^i)$ is the battery-delivered energy. Let $V_2^i(\pi^i)$ be the corresponding value function, defined as in equation 3. We then impose the local constraint

$$V_2^i(\pi^i) = 0,$$

which ensures that, in expectation, all of agent $i$'s demand is met over the long run. Since the constraint is local, it only affects the optimization problem of agent $i$. The training of the policy is performed following the state augmented procedure described in Section 4.1 and the updates of the global constraint and the consensus multipliers are performed as shown in Algorithm 1. For the handling of the local constraint we just add another term to the Lagrangian which only needs the addition of the following update

$$\nu_{k+1}^i = \nu_k^i - \eta \left( d_k^i - e_{\text{grid}}(s_k^i, a_k^i) - e_{\text{bat}}(s_k^i, a_k^i) \right), \tag{20}$$

with step size $\eta$. Energy prices, demand, and solar generation data come from City Learn (Vázquez-Canteli et al., 2020; 2019; Lab, 2024). Since this constraint is purely local, $\nu^i$ does not participate in the consensus mechanism: each $\nu^i$ is updated independently using agent $i$'s own unmet demand signal. Consequently, the local constraint does not affect the consensus analysis in Theorem 5.1 or Proposition 5.4, which concern only the global multiplier $\lambda$.

## 7    Experimental Results

We test our method[3] on the network configurations in Figure 1, which vary in connectivity and demand diversity. Less-connected networks challenge consensus, while heterogeneous demands create problems that require coordination to solve. We focus on the configuration in Figure 1c: two weakly connected groups where one agent in each has double the demand of others. Using PPO (Schulman et al., 2017) on the Farama Gymnasium framework (Towers et al., 2024), we train just two policies—one for normal demand, one for double demand—demonstrating the efficiency of single-agent training with multi-agent execution. Individual Lagrange multipliers ($\lambda^i$) enable coordination during execution, with consensus being critical for linking training to execution and ensuring constraint satisfaction. Training uses $10^6$ PPO timesteps per agent

---

[3]Execution experiments were carried out on a MacBook Pro M3 with 8 GB RAM. Training benchmarks were measured on a 16-core CPU with 21 GB RAM.

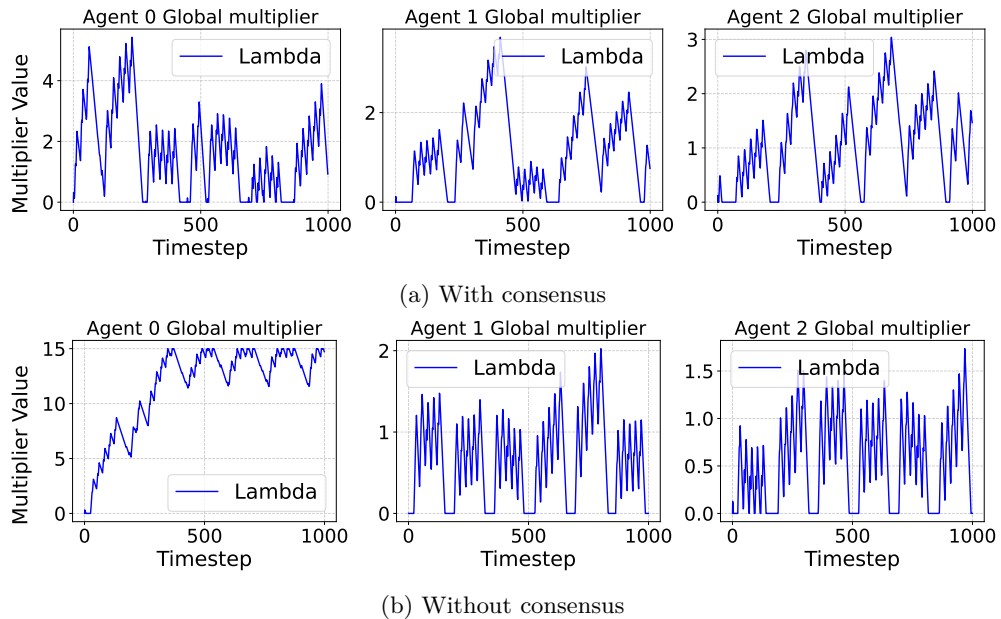

(a) With consensus

(b) Without consensus

Figure 2: Evolution of the global multipliers $\lambda^i$ for the first three agents during execution.

type, in episodes of 80 timesteps (about 3 days of demand), with multipliers sampled from $\lambda \in [0, 15]$ and $\nu \in [-25, 25]$. The dataset contains 3,000 hours of data with random episode starting points.

All multiplier step sizes are set to $\alpha = \epsilon = \eta = 0.01$, and we use $\mathscr{L} = 1$ consensus round per timestep. The step sizes were selected to balance convergence speed against oscillation amplitude; the consensus bound in Theorem 5.1 shows that the asymptotic error scales as $\alpha\,\sigma/(1 - \rho)$, so smaller $\alpha$ reduces steady-state error at the cost of slower adaptation. A single consensus round ($\mathscr{L} = 1$) is sufficient because the spectral gap $1 - \rho$ of our communication graphs is large enough to keep the consensus error within acceptable bounds (Section 7.1).

We focus on smart grid management as it naturally fits our structural assumptions while remaining complex enough to demonstrate consensus necessity. We validate the approach through: (i) ablations demonstrating the necessity of both consensus and state augmentation (Sections 7.1–7.2), (ii) comparison against a centralized oracle establishing that distributed consensus incurs negligible performance loss (Section 7.3), and (iii) scaling to 1,000 agents, far beyond CTDE capabilities (Section 7.4).

## 7.1 Consensus Necessity

We set the constraint $c$ to 27% of peak demand (challenging yet feasible). Agents run for 3,000 timesteps with continuous multiplier updates (Algorithm 1). To demonstrate coordination importance, we compare two variants: with *consensus*, agents exchange multipliers $\lambda^i$ with neighbors and average them (lines 8–13 in Algorithm 1); without *consensus*, agents only perform local gradient updates without coordination. While both maintain grid consumption below the threshold (Figure 3a), the no-consensus version achieves this by indefinitely postponing demand rather than finding a true solution.

Testing four constraint levels $c \in \{0.2, 0.3, 0.4, 0.5\}$, we find that consensus achieves stable unmet demand for feasible cases ($c \geq 0.3$), with lower constraints allowing more grid usage as expected (Figure 3b). At $c = 0.2$, even consensus cannot find a solution. Without consensus, the problem becomes infeasible even for moderate constraints like $c \in \{0.3, 0.4\}$ (Figure 4b). Crucially, at our target $c = 27\%$, the no-consensus version fails to solve the problem despite meeting grid constraints—its multipliers never converge (Figure 2b), preventing optimal solution discovery.

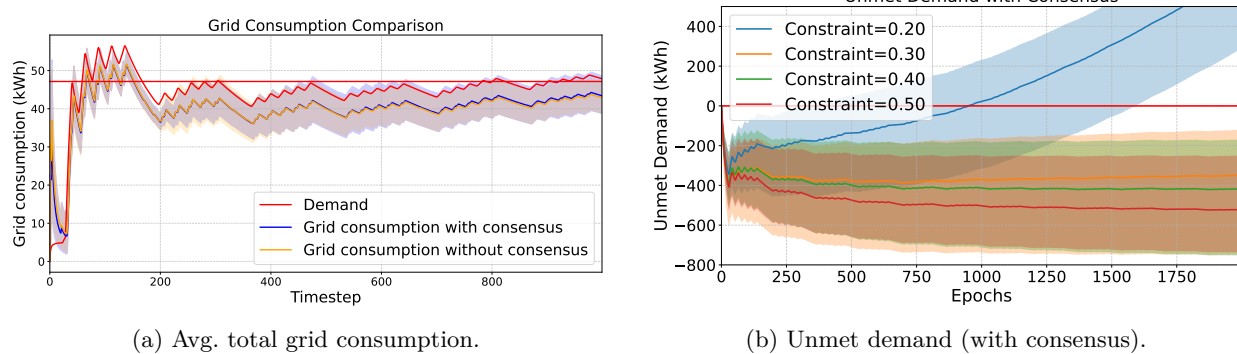

(a) Avg. total grid consumption.

(b) Unmet demand (with consensus).

Figure 3: Cumulative unmet demand with consensus. Stable negative values (c=0.3, 0.4) indicate proactive load shifting; stable zero indicates exact demand matching (no control); unbounded positive growth (c=0.2) indicates infeasibility.

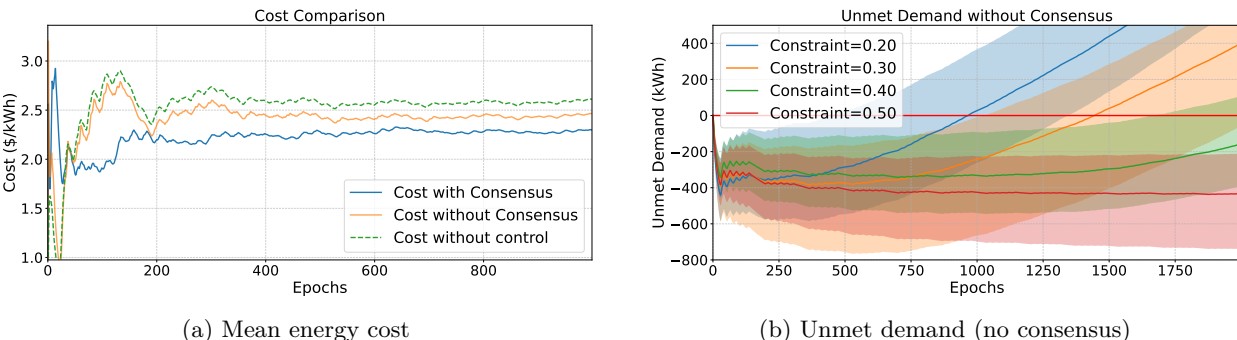

(a) Mean energy cost

(b) Unmet demand (no consensus)

Figure 4: Cost and demand-satisfaction trajectories. The dashed horizontal line in (b) marks zero unmet demand.

The absence of consensus produces both higher operating cost (Figure 4a) and continued growth of deferred demand (Figure 4b). These complementary views underline the practical value of the lightweight neighbor-to-neighbor communication adopted in Algorithm 1. Figure 2a shows that exchanging $\lambda^i$ with immediate neighbors causes convergence to the same value, thereby satisfying the global grid-consumption constraint. Without this exchange (Figure 2b), the two high-demand agents push their multipliers to the hard cap of 15, signaling that dual ascent has saturated before a feasible primal solution was found.

*Remark* 7.1 (Interpreting Unmet Demand). The cumulative unmet demand $\sum_t r_2^i$ can be positive (demand postponed) or negative (demand met early through proactive battery use). In demand response, *both directions* represent valid load shifting. The key indicator of solution quality is *stability*: a stable cumulative value (whether positive or negative) indicates the system found a feasible equilibrium, while unbounded growth indicates failure. In Figure 3b, the stable negative values for $c \in \{0.3, 0.4\}$ show agents proactively using stored energy when the grid constraint is tight,exactly the intended demand response behavior. Only $c = 0.2$ exhibits unbounded positive growth, confirming infeasibility.

## 7.2 Ablation: State Augmentation and Comparison with Baselines

Training policies with fixed Lagrange multipliers,rather than conditioning on $\lambda$ as input (state augmentation),is equivalent to standard Lagrangian approaches that Calvo-Fullana et al. (2024) prove can fail to produce feasible policies. To empirically validate this and demonstrate the necessity of state augmentation, we trained a grid of Independent PPO (IPPO) agents, one for every fixed $(\lambda, \nu) \in [0, 15] \times [-20, 8]$ (a subset of the full training sweep $\lambda \in [0, 15] \times \nu \in [-25, 25]$ retained for this feasibility analysis), yielding 414 models. Each model represents a policy trained *without* state augmentation: the multipliers are fixed during training

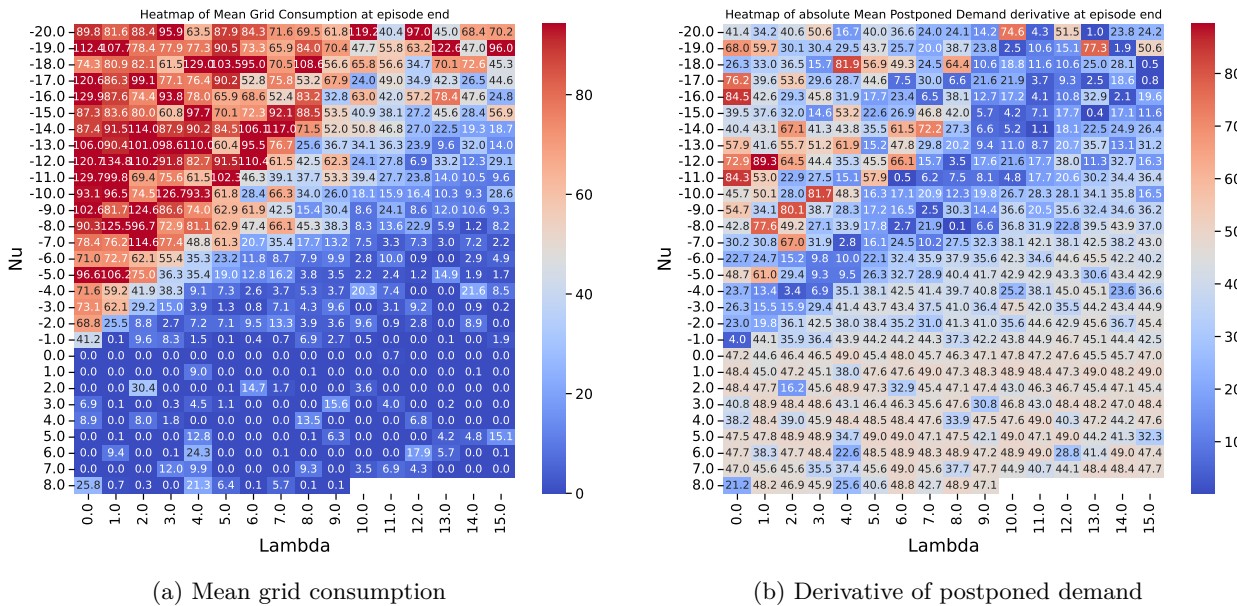

(a) Mean grid consumption

(b) Derivative of postponed demand

Figure 5: Ablation: 414 IPPO agents trained with fixed $(\lambda, \nu)$ (without state augmentation). Only 7 yield feasible behavior. Blank cells indicate diverging runs.

Table 2: The 7 fixed multiplier pairs (of 414 tested) yielding stable behavior, demonstrating the brittleness of non-augmented methods.

| | **Configuration** | | | | | | |
|---|---|---|---|---|---|---|---|
| $\lambda$ | 6 | 8 | 11 | 13 | 13 | 15 | 15 |
| $\nu$ | $-11$ | $-8$ | $-14$ | $-15$ | $-20$ | $-17$ | $-18$ |

and cannot adapt during execution. Figure 5 shows the performance for every $(\lambda, \nu)$ pair, measuring (a) mean grid consumption and (b) absolute rate of change of cumulative postponed demand. Only 7 of 414 configurations ($< 2\%$) satisfy both the 27% grid-consumption limit and prevent postponed demand from diverging (Table 2). This extreme sensitivity confirms the theoretical fragility of non-augmented methods. In contrast, our state-augmented approach trains a single policy that generalizes across all $\lambda$ values, with consensus dynamically finding the correct multiplier during execution.

**Comparison with MARL Baselines.** From the seven feasible pairs we selected $(\lambda^\star, \nu^\star) = (8, -8)$, giving baselines their best chance, and trained four multi-agent methods: MAPPO, MADDPG, MASAC, and ISAC. Each baseline ran with these hand-tuned fixed penalty weights, whereas our method adapted multipliers online via consensus. For every algorithm we executed 10 roll-outs and recorded (i) the trajectory closest to satisfying the 27% grid threshold and (ii) the mean total cost. Figure 6 shows that only MAPPO and ISAC keep grid consumption near the limit, while MASAC and MADDPG overshoot. Even these "successful" baselines closely match the operating cost of our decentralized multiplier-adaptive method, which achieves 0% infeasible roll-outs across all seeds. Despite their hand-tuned advantage from exhaustive search, the baselines still violate constraints, highlighting the fundamental limitation of fixed-multiplier approaches.

All four baselines rely on centralized components; MAPPO, MASAC and ISAC use a joint critic, while MADDPG conditions each critic on the full joint action.[4] Consequently, their computational and memory costs explode as the population grows, limiting practical use to a few dozen agents. Our method keeps both

---

[4]The policies may execute decentralized actions at test time, but training still scales at least quadratically with the number of agents because the critics ingest the joint state–action tuple.

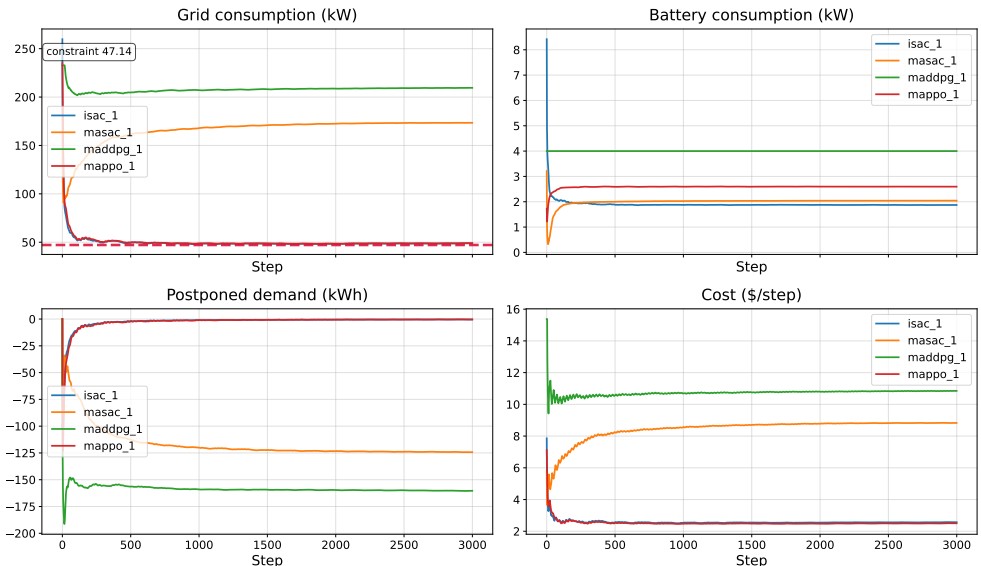

Figure 6: Average performance of state-of-the-art MARL baselines. The grey dashed line marks the grid-consumption constraint (27 % of peak demand).

training and execution fully decentralized, needs only one policy per agent type, and scales linearly in the number of agents (Figure 8a), giving it a clear advantage for large systems.

The related work discusses constrained MARL methods such as Safe Dec-PG (Lu et al., 2021), MACPO (Gu et al., 2021), and Scal-MAPPO-L (Zhang et al., 2024). These methods address different problem settings (coupled dynamics, per-step or cumulative safety constraints) and are not directly applicable to our separable-dynamics, average-constraint formulation. Rather than adapting an external method to our setting, we compare against a *centralized oracle*: a single global $\lambda$ updated with perfect knowledge of the aggregate constraint signal at every timestep. This is the strongest possible baseline under our formulation, as it represents the performance achievable with unlimited communication. Any gap between our distributed method and this oracle is precisely the cost of decentralization.

### 7.3 Centralized Oracle Comparison

To quantify the cost of decentralization, we compare against a *centralized oracle* that replaces the distributed consensus mechanism with a single global multiplier updated from the true average constraint satisfaction across all agents: $\lambda_{k+1} = [\lambda_k - \alpha \, (c/N - \frac{1}{N} \sum_{i=1}^{N} V_{1,k}^i)]_+$. This oracle has perfect global information at every timestep, an assumption that is unrealistic in practice but provides a best-case reference. Both methods use the same state-augmented policies trained offline; the only difference is how $\lambda$ is coordinated during execution.

Table 3 reports results over 10 identical random seeds. The distributed method matches the centralized oracle in all operational metrics: grid consumption is very close ($44.89 \pm 0.01\,\text{kWh}$ vs. the $47.14\,\text{kWh}$ threshold), energy cost differs by $+0.086\% \pm 0.075\%$ (distributed is marginally more expensive than the oracle, well within the $< 0.2\%$ regime), and both achieve negative cumulative unmet demand indicating proactive demand fulfillment. The constraint is satisfied in 100% of runs for both methods.

The only notable difference is in the converged dual variable: $\bar{\lambda} = 1.49 \pm 0.01$ (distributed) versus $2.11 \pm 0.01$ (centralized). Despite this difference in the multiplier values, the state-augmented policies produce operationally equivalent behavior. This demonstrates that the policies are robust to variation in $\lambda$, consistent with the Lipschitz sensitivity assumption (Assumption 5.2): the constraint value function changes smoothly with the multiplier, so moderate differences in $\lambda$ do not significantly affect constraint satisfaction or cost.

Table 3: Centralized oracle vs. distributed consensus ($N = 7$, $c = 0.27$, $T = 3000$, 10 paired seeds). Both methods use the same trained policies; the only difference is the coordination mechanism.

| Metric | Distributed | Centralized | Paired diff. |
|---|---|---|---|
| Grid consumption (kWh) | $44.887 \pm 0.009$ | $44.888 \pm 0.008$ | $-0.0013 \pm 0.0038$ |
| Constraint satisfied | $100\%$ | $100\%$ | — |
| Energy cost (\$) | $872{,}977.79 \pm 356.63$ | $872{,}228.72 \pm 568.16$ | $+749.07 \pm 650.02$ |
| Relative cost gap (\%) | — | — | $+0.086 \pm 0.075$ |
| Unmet demand (kWh) | $-157.78 \pm 218.41$ | $-108.41 \pm 216.21$ | $-49.37 \pm 4.27$ |

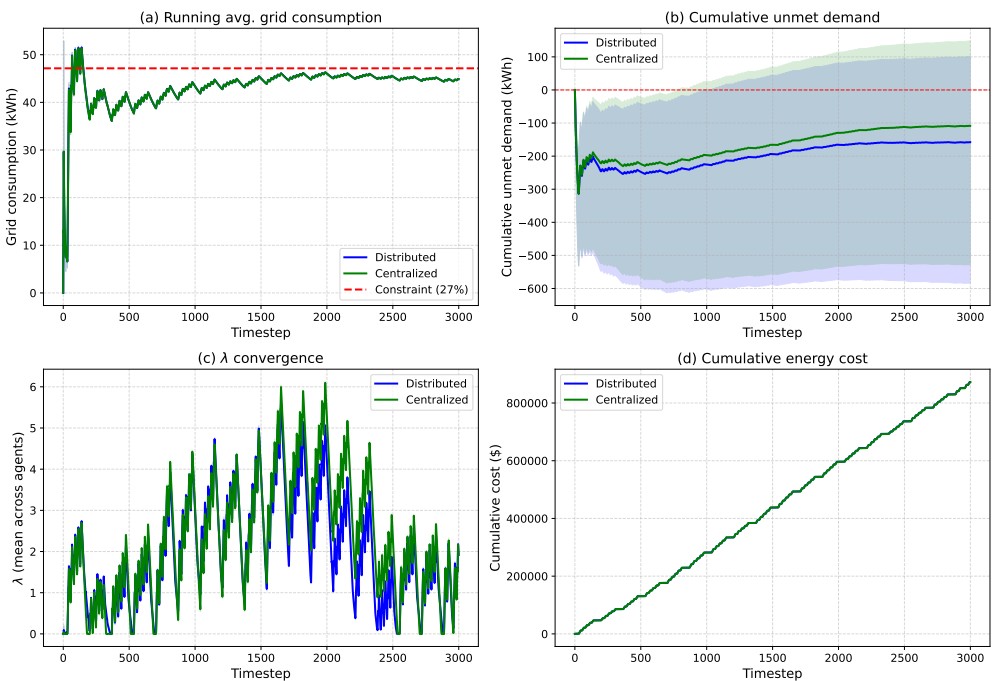

Figure 7: Centralized oracle vs. distributed consensus ($N = 7$, $c = 0.27$, 10 seeds). Both satisfy the constraint and achieve equivalent cost and demand fulfillment.

The empirical Lipschitz constant, estimated from policy evaluations over $\lambda \in [0, 15]$ (Appendix B.2, Figure B.1), is $\hat{L}_V = 8.44$ kWh per unit $\lambda$ and $\hat{L}_0 = 39.24$ \$ per unit $\lambda$. For the 7-agent configuration with $\mathscr{L} = 1$, the consensus error term in Corollary 5.5 gives $L_V \sqrt{N} \delta \approx 22.3\, \delta$ kWh. The observed constraint margin $(47.14 - 44.89 = 2.25$ kWh, Table 3) is well within this bound for the empirical consensus errors, showing that the theoretical guarantees of Proposition 5.4 are well grounded.

Figure 7 shows the execution dynamics. Both methods converge to the same grid-consumption level below the constraint threshold, with the distributed approach achieving slightly more proactive demand fulfillment (more negative cumulative unmet demand) due to its marginally more conservative operation.

## 7.4 Scalability Study

To validate scalability claims, we tested our method on systems with 10, 100, 500, and 1000 agents. Figure 8a confirms the *linear* execution-time scaling predicted by our decentralized design, while Figure 8b demonstrates that all agents converge to a common multiplier irrespective of population size. This scalability to 1000

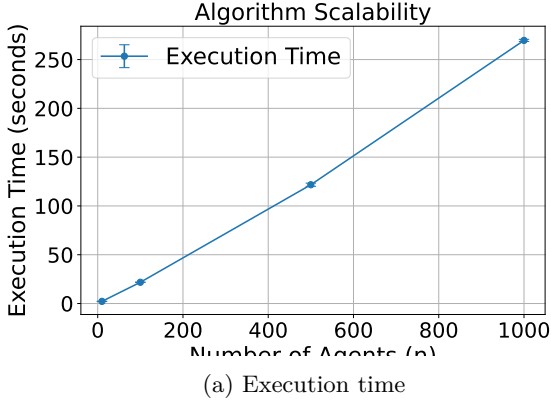

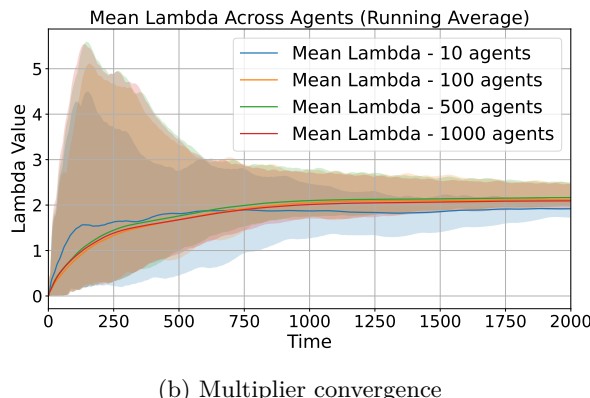

(a) Execution time

(b) Multiplier convergence

Figure 8: Scalability of the execution phase. (a) Wall-clock execution time versus agent count. (b) Running mean of $\lambda^i$ for systems of 10, 100, 500, and 1000 agents.

agents far exceeds the capabilities of CTDE-based methods, which are typically limited to a few dozen agents due to their centralized training components.

Because our method trains a single policy per agent type, not per agent, training cost is $O(1)$ in the number of agents $N$. The same two policies (one per building type, $10^6$ PPO timesteps each) serve all configurations from 7 to 1,000 agents. Training required $26.7$ minutes total ($13.2 + 13.5\,\mathrm{min}$) on a 16-core CPU with $\sim 5\,\mathrm{GB}$ peak memory. In contrast, CTDE methods require retraining whenever $N$ changes, as the centralized critic takes the joint state–action space. The combination of $O(1)$ training and linear execution scaling is what enables the orders-of-magnitude improvement over existing approaches.

## 8    Conclusion

We presented a distributed approach to constrained MARL that combines state-augmented policy learning with consensus-based coordination over dual variables. Our experiments demonstrate that the dual-variable consensus mechanism is what makes independently-trained policies collectively feasible: without it, agents resort to degenerate solutions despite individually satisfying their local objectives. The comparison with a centralized oracle in Section 7.3 shows that, on the smart-grid configurations evaluated, distributed consensus closes the cost gap to $+0.086\% \pm 0.075\%$ (within the $< 0.2\%$ regime), while training cost is $O(1)$ in the number of agents and execution scales linearly, reaching 1,000 agents where CTDE methods are limited to tens. We emphasize that this near-equivalence with the centralized oracle is an empirical observation in a specific testbed, not a general optimality claim. The analysis of Appendix A.1 bounds the consensus error of the global multiplier $\lambda$ and translates this bound into a feasibility margin through Proposition 5.4, but does not yield a full primal optimality guarantee. The local-constraint multiplier $\nu$ is updated by an analogous primal-dual scheme and is well-behaved in our experiments; however, its convergence is not covered by Theorem 5.1, and we treat it as an empirical extension of the global-constraint theory rather than a setting in which the formal guarantees apply.

The separable-dynamics and summable-rewards assumptions are restrictive, and identifying the class of problems for which our framework is appropriate is therefore important. Our method is naturally suited to infrastructure-management settings in which agents have physically decoupled local dynamics and interact only through a shared resource budget. Smart-grid demand response (studied here), distributed electric-vehicle charging, traffic-signal coordination at lightly-coupled intersections, and water- or heating-network management all share this structure: each node manages a local controllable resource, and the global constraint aggregates linearly across nodes. Conversely, settings in which agent dynamics are physically coupled, such as multi-robot manipulation, cooperative navigation in a shared workspace, or pursuit-evasion in a joint state space, violate Assumption 3.2 and lie outside the regime to which our analysis applies. For such problems the

consensus error bound of Theorem 5.1 no longer guarantees that aligned multipliers produce a coherent joint policy, and a different coordination mechanism is required.

The empirical evidence we present is encouraging but comes predominantly from one application domain, and broader validation across the application classes identified above remains a priority for follow-up work. Several further directions are open. First, extending the framework to time-varying constraints would broaden applicability to settings where resource limits fluctuate, such as renewable-energy availability or congestion-dependent capacities. Second, heterogeneous and time-evolving communication graphs would better model realistic infrastructure networks. Third, hybrid approaches that combine our lightweight consensus mechanism with more expressive coordination, for example sparse joint critics on small clusters of strongly coupled agents, could address problems with partially-coupled dynamics and bridge the gap between full separability and the general coupled setting. Fourth, a theoretical treatment of the local-constraint multiplier $\nu$ remains open; establishing convergence guarantees analogous to Theorem 5.1 for the joint $(\lambda, \nu)$ update is a natural next step.

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

# A   Theoretical Analysis

## A.1   Convergence of the Consensus Algorithm

In this section, we provide a rigorous analysis of the convergence properties of the consensus algorithm employed in our distributed optimization framework. The convergence properties of consensus algorithms over networks are well-studied in the literature (Olfati-Saber & Murray, 2004; Olfati-Saber et al., 2007; Xiao & Boyd, 2003). Our analysis follows standard techniques in distributed optimization and consensus algorithms, as well as properties of graph Laplacians and their spectra (Chung, 1997). Specifically, we leverage results from spectral graph theory and matrix analysis to establish the exponential convergence of our algorithm. We examine how the local dual variables $\lambda^i$ converge to a consensus value, ensuring coordination among agents while satisfying global constraints.

### A.1.1   Consensus Update Rule

The consensus update for agent $i$ at iteration $\ell$ can be written in a standard form for consensus algorithms:

$$
\begin{aligned}
\lambda_{\ell+1}^i &= \lambda_\ell^i - \epsilon \left( \lambda_\ell^i - \frac{1}{|\mathcal{N}^i|} \sum_{j \in \mathcal{N}^i} \lambda_\ell^j \right), \\
&= \lambda_\ell^i - \epsilon \left( \frac{1}{|\mathcal{N}^i|} \sum_{j \in \mathcal{N}^i} (\lambda_\ell^i - \lambda_\ell^j) \right), \\
&= \lambda_\ell^i - \epsilon \sum_{j \in \mathcal{N}^i} \frac{1}{|\mathcal{N}^i|} \left( \lambda_\ell^i - \lambda_\ell^j \right).
\end{aligned}
\tag{A.1}
$$

where $\epsilon > 0$ is the consensus step size. This update rule adjusts each agent's dual variable towards the average of its neighbors' dual variables.

### A.1.2   Matrix Formulation

We consider a communication network among the agents, given by an undirected graph $G = (V, E)$, where $V$ is the set of vertices (agents) and $E \subset V \times V$ is the set of edges (communication links between agents). The neighborhood of a node $i \in V$, denoted by $\mathcal{N}^i$, is the set of nodes that are directly connected to node $i$ by an edge; i.e., $\mathcal{N}^i = \{ j \in V \mid (i,j) \in E \}$.

We aim to express the collective updates in matrix form to facilitate the convergence analysis. To do this, we first define the necessary matrices and vectors.

Let $\lambda_\ell = [\lambda_\ell^1, \lambda_\ell^2, \ldots, \lambda_\ell^N]^T \in \mathbb{R}^N$ be the global vector of local dual variables at iteration $\ell$, and let $\mathbf{1} \in \mathbb{R}^N$ be a vector of ones. We denote by $A \in \mathbb{R}^{N \times N}$ the adjacency matrix of the graph, where

$$
A(i,j) \;=\; \begin{cases} 1, & \text{if } (i,j) \in E, \\ 0, & \text{otherwise,} \end{cases}
\tag{A.2}
$$

and by $D \in \mathbb{R}^{N \times N}$ the diagonal degree matrix with $D(i,i) = |\mathcal{N}^i|$. We define the (unnormalized) graph Laplacian as $L = D - A$, and the random-walk normalized Laplacian as $L^{\mathrm{rw}} = D^{-1}L = I - D^{-1}A$.

Then the update of $\lambda_{\ell+1}$ in vector form is:

$$
\begin{aligned}
\lambda_{\ell+1} &= \lambda_\ell - \epsilon \left( \lambda_\ell - D^{-1}A\lambda_\ell \right), \\
&= \lambda_\ell - \epsilon L^{\mathrm{rw}} \lambda_\ell, \\
&= P\lambda_\ell,
\end{aligned}
\tag{A.3}
$$

where $P = I - \epsilon L^{\mathrm{rw}}$ is the *Perron matrix*, and $I$ is the identity matrix. The graph Laplacian $L^{\mathrm{rw}}$ captures the connectivity of the communication network among agents.

### A.1.3 Assumptions for Convergence

To analyze the convergence of the consensus algorithm, we make the following assumptions:

**Assumption A.1** (Connected Graph). The communication graph $G = (V, E)$ is undirected and connected; that is, there exists a path between any pair of agents.

**Assumption A.2** (Step Size). The consensus step size $\epsilon$ satisfies $0 < \epsilon < 1$, ensuring that $P$ remains a stochastic matrix with non-negative entries.

Assumption A.1 ensures that information can propagate through the network, which is necessary for achieving global consensus. Assumption A.2 provides a bound on the step size to guarantee convergence.

### A.1.4 Convergence Analysis

We analyze the convergence of the consensus algorithm by examining the properties of the Perron matrix $P$.

**Lemma A.3** (Properties of the Perron Matrix). *Under Assumptions A.1 and A.2, the Perron matrix*
$P = I - \epsilon L^{rw}$ *satisfies the following properties:*
*(a) $P$ is row-stochastic and irreducible.*
*(b) The eigenvalues of $P$ are $\nu_i = 1 - \epsilon \Lambda_i$, where $\Lambda_i$ are the eigenvalues of the Laplacian $L^{rw}$.*
*(c) All eigenvalues of $P$ satisfy $|\nu_i| \leq 1$, the eigenvalue $\nu_1 = 1$ has algebraic multiplicity one, and all other eigenvalues satisfy $|\nu_i| < 1$.*

*Proof.* **(a) Row-Stochasticity and Irreducibility:** The elements of $P$ are given by

$$P(i,j) = \begin{cases} 1 - \epsilon, & \text{if } i = j, \\ \frac{\epsilon}{|\mathcal{N}^i|}, & \text{if } (i,j) \in E, \\ 0, & \text{otherwise.} \end{cases}$$

For each row $i$, the sum of the entries is

$$\begin{aligned} \sum_{j=1}^{N} P(i,j) &= P(i,i) + \sum_{j \in \mathcal{N}^i} P(i,j) \\ &= (1 - \epsilon) + \sum_{j \in \mathcal{N}^i} \frac{\epsilon}{|\mathcal{N}^i|} \\ &= (1 - \epsilon) + \epsilon \frac{|\mathcal{N}^i|}{|\mathcal{N}^i|} \\ &= (1 - \epsilon) + \epsilon = 1. \end{aligned}$$

Thus, $P$ is row-stochastic. Since the graph $G$ is connected (Assumption A.1), and $P$ is non-negative, it follows that $P$ is irreducible.

**(b) Eigenvalues of $P$:** Let $\Lambda_i$ be the eigenvalues of $L^{\mathrm{rw}}$ with corresponding eigenvectors $v_i$. Then,

$$L^{\mathrm{rw}} v_i = \Lambda_i v_i.$$

Therefore,

$$Pv_i = (I - \epsilon L^{\mathrm{rw}}) v_i = v_i - \epsilon L^{\mathrm{rw}} v_i = v_i - \epsilon \Lambda_i v_i = (1 - \epsilon \Lambda_i) v_i.$$

Thus, the eigenvalues of $P$ are $\nu_i = 1 - \epsilon \Lambda_i$.

**(c) Eigenvalues within $[-1, 1]$:** Since the random-walk Laplacian $L^{\mathrm{rw}}$ is similar to the symmetric normalized Laplacian

$$L^{\mathrm{sym}} = D^{-\frac{1}{2}} L D^{-\frac{1}{2}}$$

via

$$L^{\mathrm{sym}} = D^{\frac{1}{2}} L^{\mathrm{rw}} D^{-\frac{1}{2}},$$

they share the same set of eigenvalues $\{\Lambda_i\}$.

To see this more explicitly, let $\Lambda_i$ and $v_i$ be an eigenvalue–eigenvector pair of $L^{\mathrm{sym}}$, i.e.,

$$L^{\mathrm{sym}}\, v_i \;=\; \Lambda_i\, v_i \quad \Longleftrightarrow \quad \left(I \;-\; D^{-\frac{1}{2}} A\, D^{-\frac{1}{2}}\right) v_i \;=\; \Lambda_i\, v_i.$$

Pre-multiplying both sides by $D^{-\frac{1}{2}}$ gives

$$\left(D^{-\frac{1}{2}} \;-\; D^{-1}A\, D^{-\frac{1}{2}}\right) v_i \;=\; \Lambda_i\, D^{-\frac{1}{2}}\, v_i \quad \Longleftrightarrow \quad \left(I \;-\; D^{-1}A\right) D^{-\frac{1}{2}}\, v_i \;=\; \Lambda_i\, D^{-\frac{1}{2}}\, v_i.$$

Recalling that $L^{\mathrm{rw}} = I - D^{-1}A$, it follows that

$$L^{\mathrm{rw}}\left(D^{-\frac{1}{2}}\, v_i\right) \;=\; \Lambda_i\left(D^{-\frac{1}{2}}\, v_i\right).$$

Hence, if $(\Lambda_i, v_i)$ is an eigenvalue–eigenvector pair of $L^{\mathrm{sym}}$, then the same $\Lambda_i$ and $D^{-\frac{1}{2}} v_i$ form an eigenvalue–eigenvector pair of $L^{\mathrm{rw}}$. Therefore, both matrices share the same eigenvalues. We can establish the following facts:

1. **Real symmetry and positive semidefiniteness:** The matrix $L^{\mathrm{sym}}$ is real symmetric (since $L$ is symmetric and $D^{-1/2}$ is diagonal). Then, $L^{\mathrm{sym}}$ is diagonalizable, and its eigenvalues are real. Standard results in spectral graph theory further show $L^{\mathrm{sym}}$ is positive semidefinite, implying its eigenvalues are nonnegative (Chung, 1997; Godsil & Royle, 2001).

2. **Eigenvalues in $[0,2]$:** From classical bounds on the spectrum of $L^{\mathrm{sym}}$ (e.g., using the structure of the degree and adjacency matrices), one obtains

$$0 = \Lambda_1 \le \Lambda_2 \le \cdots \le \Lambda_N \le 2 \quad \text{(Horn \& Johnson, 2012; Mohar et al., 1991)}.$$

   The eigenvalues of $L^{\mathrm{rw}}$ also lie in $[0,2]$.

Because $0 \le \Lambda_i \le 2$ and $0 < \epsilon < 1$, we have

$$|\nu_i| = |1 - \epsilon\Lambda_i| \le 1,$$

Since $G$ is connected, the multiplicity of the zero eigenvalue of $L^{\mathrm{rw}}$ is one, so the eigenvalue $\nu_1 = 1$ of $P$ has algebraic multiplicity one. Since there are no complex eigenvalues ($L^{sym}$ is real and symmetric), all other eigenvalues satisfy $|\nu_i| < 1$ for $i \ge 2$. Hence, all eigenvalues of $P$ lie in $[-1,1]$.

This ensures that the spectral radius of $P$ is $\rho(P) = 1$, and the convergence of the consensus algorithm is governed by the second-largest eigenvalue in magnitude, which is less than 1. □

## A.2 Global Consensus Error

We analyze the convergence of the consensus algorithm by first establishing the value to which it converges, and then proving the rate of convergence.

### A.2.1 Consensus Value

**Lemma A.4** (Consensus Value). *Under Assumption A.1, the consensus algorithm converges to a weighted average of the initial dual variables. Specifically, for any initial vector $\lambda_0 \in \mathbb{R}^N$,*

$$\lim_{\ell \to \infty} \lambda_\ell = \hat{\lambda}\mathbf{1},$$

*where*

$$\hat{\lambda} = \sum_{i=1}^{N} w^i \lambda_0^i,$$

*and the weights $w^i$ are given by*

$$w^i = \frac{|\mathcal{N}^i|}{\sum_{j=1}^N |\mathcal{N}^j|}.$$

*Proof.* By the Perron-Frobenius theorem (Horn & Johnson, 2012), since $P$ is a primitive nonnegative matrix, it satisfies

$$\lim_{\ell \to \infty} P^\ell = \mathbf{v}_1 \mathbf{w}_1^\top,$$

where $\mathbf{v}_1 = \mathbf{1}$ is the right eigenvector corresponding to the eigenvalue 1, and $\mathbf{w}_1$ is the unique left eigenvector satisfying $\mathbf{w}_1^\top P = \mathbf{w}_1^\top$ with $\mathbf{v}_1^\top \mathbf{w}_1 = 1$.

The consensus iteration is given by

$$\lambda_\ell = P^\ell \lambda_0. \tag{A.4}$$

Taking the limit as $\ell \to \infty$,

$$\lim_{\ell \to \infty} \lambda_\ell = \lim_{\ell \to \infty} P^\ell \lambda_0 = \mathbf{v}_1 \mathbf{w}_1^\top \lambda_0 = \mathbf{1}(\mathbf{w}_1^\top \lambda_0) = \hat{\lambda}\mathbf{1}.$$

This shows that all agents' dual variables converge to the scalar $\hat{\lambda}$, which is a weighted average of the initial values.

To explicitly determine $\mathbf{w}_1$, consider the transition matrix $P^{\mathrm{rw}} = D^{-1}A$, associated with the random walk normalized Laplacian $L^{rw}$ where $A$ is the adjacency matrix and $D$ is the degree matrix. For an undirected graph, $P^{\mathrm{rw}}$ satisfies the detailed balance condition (Levin et al., 2009):

$$w^i P_{ij}^{\mathrm{rw}} = w^j P_{ji}^{\mathrm{rw}}.$$

Substituting $P^{rw}(i,j) = \frac{A(i,j)}{|\mathcal{N}^i|}$ and $P^{rw}(j,i) = \frac{A(j,i)}{|\mathcal{N}^j|}$, and since $A(i,j) = A(j,i)$ for undirected graphs, we obtain

$$\frac{w^i}{|\mathcal{N}^i|} = \frac{w^j}{|\mathcal{N}^j|} = c,$$
$$w^i = c|\mathcal{N}^i|$$

Since $\sum_{i=1}^N w^i = 1$, this implies that

$$c = \frac{1}{\sum_{i=1}^N |\mathcal{N}^i|},$$

Therefore, the consensus value is

$$\hat{\lambda} = \sum_{i=1}^N w^i \lambda_0^i = \frac{\sum_{i=1}^N |\mathcal{N}^i| \lambda_0^i}{\sum_{i=1}^N |\mathcal{N}^i|},$$

which is the degree-weighted average of the initial dual variables. $\qquad\square$

### A.2.2 Convergence Rate

We now establish the exponential convergence rate to the consensus value $\hat{\lambda}$.

**Theorem A.5** (Exponential Convergence to Consensus)**.** *Under Assumptions A.1 and A.2, the consensus algorithm converges exponentially fast to $\hat{\lambda}\mathbf{1}$. Specifically, for any initial vector $\lambda_0 \in \mathbb{R}^N$,*

$$\|\lambda_\ell - \hat{\lambda}\mathbf{1}\| \le C\rho^\ell \|\lambda_0 - \hat{\lambda}\mathbf{1}\|, \tag{A.5}$$

*where:*

- $\rho = \max_{i \ge 2} |\nu_i| < 1$ *where $\nu_i$ are the eigenvalues of $P$.*

- $C = \kappa(V) = \|V\|\|V^{-1}\|$ *is the condition number of the eigenvector matrix* $V$.

*Proof.* Define the error vector at iteration $\ell$ as:

$$e_\ell = \lambda_\ell - \hat{\lambda}\mathbf{1}.$$

From Lemma A.4, we have $\lim_{\ell\to\infty} e_\ell = \mathbf{0}$.

The consensus update rule is:

$$\lambda_{\ell+1} = P\lambda_\ell,$$

which implies:

$$e_{\ell+1} = Pe_\ell.$$

Iterating this, we obtain:

$$e_\ell = P^\ell e_0.$$

Since $P$ is diagonalizable, we can express it as:

$$P = V\Gamma V^{-1},$$

where:

- $V$ is the matrix of right eigenvectors of $P$.

- $\Gamma = \mathrm{diag}(\nu_1, \nu_2, \ldots, \nu_N)$ contains the eigenvalues of $P$, with $\nu_i = 1 - \epsilon\Lambda_i$.

Substituting into the error expression:

$$e_\ell = V\Gamma^\ell V^{-1}e_0. \tag{A.6}$$

In the degree-weighted consensus setting, for eigenvalue 1, the matrix $P$ has $\mathbf{1}$ as its right eigenvector, while its left eigenvector is $\mathbf{w}_1$. By definition, the initial error is $e_0 = \lambda_0 - \hat{\lambda}\mathbf{1}$ (where $\hat{\lambda}$ is the weighted average), we then have

$$\mathbf{w}_1^\top e_0 = \sum_{i=1}^{N} w^i(\lambda_0^i - \hat{\lambda}) = 0.$$

Thus, $e_0$ lies in the subspace orthogonal to $\mathbf{w}_1$. Since $e_\ell = P^\ell e_0$ and $\mathbf{w}_1^\top P = \mathbf{w}_1^\top$, it follows that

$$\mathbf{w}_1^\top e_\ell = \mathbf{w}_1^\top P^\ell e_0 = \mathbf{w}_1^\top e_0 = 0 \quad \text{for all } \ell.$$

Hence, the error remains in the subspace orthogonal to $\mathbf{w}_1$ at every iteration, allowing us to exclude the dominant component in the consensus convergence analysis. Thus,

$$e_\ell = V_{\mathrm{red}}\Gamma_{\mathrm{red}}^\ell V_{\mathrm{red}}^{-1}e_0,$$

where:

- $V_{\mathrm{red}}$ consists of eigenvectors corresponding to $\nu_i$ for $i \geq 2$.

- $\Gamma_{\mathrm{red}} = \mathrm{diag}(\nu_2, \nu_3, \ldots, \nu_N)$.

To bound the norm of the error, we apply the sub-multiplicative property of matrix norms:

$$\|e_\ell\| \leq \|V_{\mathrm{red}}\|\|\Gamma_{\mathrm{red}}^\ell\|\|V_{\mathrm{red}}^{-1}\|\|e_0\|.$$

Since $\|\Gamma_{\text{red}}^\ell\|_2 = \rho^\ell$, where $\rho = \max_{i \geq 2} |\nu_i|$, we have:

$$\|e_\ell\| \leq \|V\|\|V^{-1}\|\rho^\ell\|e_0\| = C\rho^\ell\|e_0\|,$$

where $C = \kappa(V) = \|V\|\|V^{-1}\|$ is the condition number of $V$.

Since $\rho < 1$, the error decays exponentially:

$$\|e_\ell\| \leq C\rho^\ell\|e_0\|,$$

confirming that the consensus algorithm converges exponentially fast to $\hat{\lambda}\mathbf{1}$. $\qquad\square$

### A.2.3 Bounding the Global Consensus Error

We aim to bound the consensus error $e_{k+1} = \lambda_{k+1} - \hat{\lambda}_{k+1}\mathbf{1}$, where $\hat{\lambda}_{k+1}$ is the weighted average of $\lambda_{k+1}^i$.

**Lemma A.6** (Consensus Error Recursion). *Under Assumptions A.1 and A.2, the magnitude of the consensus error satisfies the recursion*

$$\|e_{k+1}\| \leq \left\|P^{\mathscr{L}}\right\| \|(e_k + \alpha\Delta V_{1,k})\|, \tag{A.7}$$

*where $\Delta V_{1,k} = V_{1,k} - \hat{V}_{1,k}\mathbf{1}$ and $\hat{V}_{1,k} = \frac{\sum_{i=1}^N |\mathcal{N}^i|V_{1,k}^i}{\sum_{i=1}^N |\mathcal{N}^i|}$.*

*Proof.* From the gradient descent step,

$$\lambda_{k+\frac{1}{2}}^i = \left[\lambda_k^i - \alpha\left(\frac{c}{N} - V_{1,k}^i\right)\right]_+.$$

The weighted average is

$$\hat{\lambda}_{k+\frac{1}{2}} = \sum_{i=1}^N w^i \lambda_{k+\frac{1}{2}}^i.$$

$$= \sum_{i=1}^N w^i \left[\lambda_k^i - \alpha\left(\frac{c}{N} - V_{1,k}^i\right)\right]_+.$$

Using the non-expansive property of the projection $[\cdot]_+$, we can write the magnitude of the error after the gradient step as

$$\left\|e_{k+\frac{1}{2}}^i\right\| = \left\|\lambda_{k+\frac{1}{2}}^i - \hat{\lambda}_{k+\frac{1}{2}}\right\| \tag{A.8}$$

$$= \left\|\left[\lambda_k^i - \alpha\left(\frac{c}{N} - V_{1,k}^i\right)\right]_+ - \sum_{i=1}^N w^i \left[\lambda_k^i - \alpha\left(\frac{c}{N} - V_{1,k}^i\right)\right]_+\right\|,$$

$$\leq \left\|\lambda_k^i - \alpha\left(\frac{c}{N} - V_{1,k}^i\right) - \sum_{i=1}^N w^i \left(\lambda_k^i - \alpha\left(\frac{c}{N} - V_{1,k}^i\right)\right)\right\|,$$

$$= \left\|\lambda_k^i - \alpha\left(\frac{c}{N} - V_{1,k}^i\right) - \left(\hat{\lambda}_k - \alpha\left(\frac{c}{N} - \hat{V}_{1,k}\right)\right)\right\|,$$

$$= \left\|e_k^i + \alpha\left(V_{1,k}^i - \hat{V}_{1,k}\right)\right\|. \tag{A.9}$$

After the consensus update equation A.4,

$$\|e_{k+1}\| \leq \left\|P^{\mathscr{L}}\right\| \left\|e_{k+\frac{1}{2}}\right\|. \tag{A.10}$$

since the consensus step only affects the error term through multiplication by $P$. Substituting equation A.9 into equation A.10 and letting $V_{1,k}^i - \hat{V}_{1,k} = \Delta V_{1,k}$, we obtain equation A.7. $\qquad\square$

**Theorem A.7** (Asymptotic Bound on Consensus Error). *For the standard assumption of bounded rewards, the constraint functions $V_{1,k}^i$ are bounded such that $\|\Delta V_{1,k}\| \leq \sigma$ for some $\sigma > 0$. Then, the consensus error satisfies*

$$\lim_{k \to \infty} \|e_{k+1}\| \leq \frac{\rho^{\mathscr{L}} \alpha \sigma}{1 - \rho^{\mathscr{L}}}.$$

*where $\rho = 1 - \epsilon \Lambda_2$ as before.*

*Proof.* Using Lemma A.6 and Theorem A.5, we have

$$\|e_{k+1}\| \leq \|P^{\mathscr{L}}\| (\|e_k\| + \alpha \|\Delta V_{1,k}\|)$$
$$\leq \rho^{\mathscr{L}} \|e_k\| + \rho^{\mathscr{L}} \alpha \sigma, \tag{A.11}$$

since $\|P^{\mathscr{L}}\| = \rho^{\mathscr{L}}$ in the subspace orthogonal to $\mathbf{1}$.

Unrolling the recursion:

$$\|e_{k+1}\| \leq \rho^{\mathscr{L}} \|e_k\| + \rho \alpha \sigma$$
$$\leq \rho^{2\mathscr{L}} \|e_{k-1}\| + \rho^{2\mathscr{L}} \alpha \sigma + \rho^{\mathscr{L}} \alpha \sigma$$
$$\leq \dots$$
$$\leq \rho^{\mathscr{L}(k+1)} \|e_0\| + \rho^{\mathscr{L}} \alpha \sigma \sum_{t=0}^{k} \rho^{t\mathscr{L}}$$
$$= \rho^{\mathscr{L}(k+1)} \|e_0\| + \rho^{\mathscr{L}} \alpha \sigma \left( \frac{1 - \rho^{\mathscr{L}(k+1)}}{1 - \rho^{\mathscr{L}}} \right).$$

Taking the limit as $k \to \infty$, we obtain

$$\lim_{k \to \infty} \|e_{k+1}\| \leq \frac{\rho^{\mathscr{L}} \alpha \sigma}{1 - \rho^{\mathscr{L}}}.$$

since $\|P^{\mathscr{L}}\| = \rho^{\mathscr{L}}$ in the subspace orthogonal to $\mathbf{1}$ (the condition number $C = \kappa(V)$ from Theorem A.5 does not appear here because the error vector $e_k$ lies entirely in this subspace by construction (see the proof of Theorem A.5) so the dominant eigenvalue $\nu_1 = 1$ and its associated eigenvector direction are absent from the error dynamics). $\square$

## B Sensitivity Analysis: From Consensus Error to Feasibility

We prove Proposition 5.4 and then discuss the conditions under which the Lipschitz assumption holds.

### B.1 Proof of Proposition 5.4

*Proof.* **Part (a).** We decompose the left-hand side of equation 15 using the triangle inequality:

$$\left| \sum_{i=1}^{N} V_1^i(\hat{\pi}^i(\lambda^i)) - \sum_{i=1}^{N} V_1^i(\pi_\star^i(\bar{\lambda})) \right|$$
$$\leq \underbrace{\sum_{i=1}^{N} |V_1^i(\hat{\pi}^i(\lambda^i)) - V_1^i(\pi_\star^i(\lambda^i))|}_{(I)} + \underbrace{\sum_{i=1}^{N} |V_1^i(\pi_\star^i(\lambda^i)) - V_1^i(\pi_\star^i(\bar{\lambda}))|}_{(II)}. \tag{B.1}$$

For term (I), Assumption 5.3 yields (I) $\leq N \varepsilon_{\text{approx}}$. For term (II), Assumption 5.2 gives

$$(II) \leq L_V \sum_{i=1}^{N} |\lambda^i - \bar{\lambda}| \leq L_V \sqrt{N} \|\boldsymbol{\lambda} - \bar{\lambda} \mathbf{1}\| \leq L_V \sqrt{N} \delta, \tag{B.2}$$

where the second inequality is the Cauchy–Schwarz inequality ($\|\mathbf{x}\|_1 \le \sqrt{N}\,\|\mathbf{x}\|_2$) and the third follows from the premise $\|\boldsymbol{\lambda} - \bar{\lambda}\,\mathbf{1}\| \le \delta$. Combining gives equation 15.

**Part (b).** We write

$$
\sum_{i=1}^{N} V_1^i\big(\hat{\pi}^i(\lambda^i)\big) = \sum_{i=1}^{N} V_1^i\big(\pi_\star^i(\bar{\lambda})\big) \; + \; \left[\sum_{i=1}^{N} V_1^i\big(\hat{\pi}^i(\lambda^i)\big) - \sum_{i=1}^{N} V_1^i\big(\pi_\star^i(\bar{\lambda})\big)\right]
$$
$$
\le \sum_{i=1}^{N} V_1^i\big(\pi_\star^i(\bar{\lambda})\big) + L_V \sqrt{N}\,\delta + N\,\varepsilon_{\mathrm{approx}}, \tag{B.3}
$$

where the inequality applies the upper bound from part (a). Next, using Assumption 5.2 again,

$$
\sum_{i=1}^{N} V_1^i\big(\pi_\star^i(\bar{\lambda})\big) \; \le \; \sum_{i=1}^{N} V_1^i\big(\pi_\star^i(\lambda^\star)\big) + L_V\,N\,|\bar{\lambda} - \lambda^\star| \; \le \; c + L_V\,N\,|\bar{\lambda} - \lambda^\star|, \tag{B.4}
$$

where the final inequality uses the assumed centralized feasibility $\sum_i V_1^i(\pi_\star^i(\lambda^\star)) \le c$. Substituting equation B.4 into equation B.3 yields equation 16.

**Part (c).** The argument is identical to parts (a)–(b), replacing $V_1^i$ by $V_0^i$ and the Lipschitz constant $L_V$ by $L_0$:

$$
\left|\sum_{i=1}^{N} V_0^i\big(\hat{\pi}^i(\lambda^i)\big) - \sum_{i=1}^{N} V_0^i\big(\pi_\star^i(\lambda^\star)\big)\right| \le \underbrace{L_0 \sqrt{N}\,\delta}_{\substack{\text{consensus}\\\text{error}}} + \underbrace{L_0\,N\,|\bar{\lambda} - \lambda^\star|}_{\substack{\text{dual}\\\text{convergence}}} + \underbrace{N\,\varepsilon_{\mathrm{approx}}}_{\substack{\text{function}\\\text{approx.}}},
$$

which gives equation 17. $\qquad\square$

## B.2 Discussion of Assumptions

*Remark* B.1 (When Does Assumption 5.2 Hold?). The Lipschitz condition equation 13 captures how sensitively each agent's constraint-relevant behavior changes with the dual variable. We provide three complementary justifications.

**(i) State-augmented neural policies.** In our framework, $\lambda$ enters as an input to the neural network policy $\pi_\theta^i(a \mid s, \lambda)$. Standard neural networks with bounded weights are Lipschitz functions of their inputs (Szegedy et al., 2014; Miyato et al., 2018). Standard sensitivity results for MDPs (Kearns & Singh, 2002; Kakade & Langford, 2002) then imply that $V_1^i(\pi_\theta^i(\cdot, \lambda))$ inherits Lipschitz continuity in $\lambda$, with a constant determined by the network's spectral norm and the effective horizon. Concretely, if the policy network has Lipschitz constant $L_\pi$ with respect to $\lambda$, the transition kernel is $L_P$-Lipschitz in the total-variation sense, and the effective horizon is $H_{\mathrm{eff}}$, then $L_V = O(L_\pi L_P H_{\mathrm{eff}}(r_{\max} - r_{\min}))$.

**(ii) Envelope theorem / Danskin's theorem.** The dual function $g^i(\lambda) = \max_{\pi^i}[V_0^i(\pi^i) + \lambda(c/N - V_1^i(\pi^i))]$ is convex in $\lambda$, and by Danskin's theorem its (sub)derivative is $c/N - V_1^i(\pi_\star^i(\lambda))$ whenever the maximizer is unique. Since $g^i$ is convex with bounded (sub)gradients (by the boundedness of $V_1^i$), its derivative $c/N - V_1^i(\pi_\star^i(\lambda))$ is a monotone function of $\lambda$ with bounded range—a strong structural property supporting Assumption 5.2. For the assumption to fail, $V_1^i(\pi_\star^i(\lambda))$ would need to jump discontinuously in $\lambda$, which is precluded by the smooth parameterization of state-augmented policies.

**(iii) Empirical verification.** In practice, $L_V$ can be estimated by evaluating $V_1^i(\hat{\pi}^i(\lambda))$ on a grid of $\lambda$ values and computing the maximum finite difference: $\hat{L}_V = \max_{i,k} |V_1^i(\hat{\pi}^i(\lambda_{k+1})) - V_1^i(\hat{\pi}^i(\lambda_k))| / |\lambda_{k+1} - \lambda_k|$. Since the policies are already trained over $\lambda \in [0, 15]$, this requires only additional roll-outs at selected multiplier values. Figure B.1 shows the result for both building types over $\lambda \in [0, 15]$ with $\nu = -10$ fixed. Both curves are smooth and monotonically decreasing, consistent with the Danskin argument above. The maximum finite-difference ratios are $\hat{L}_V = 6.15\,\mathrm{kWh}/\lambda$ (building type 1, at $\lambda \in [7.5, 8.0]$) and $\hat{L}_V = 8.44\,\mathrm{kWh}/\lambda$ (building type 5, at $\lambda \in [8.5, 9.0]$). The higher sensitivity of building type 5 reflects its double demand. For the cost value function, $\hat{L}_0 = 39.24\,\$/\lambda$. We use the worst-case $\hat{L}_V = 8.44$ in all bounds.

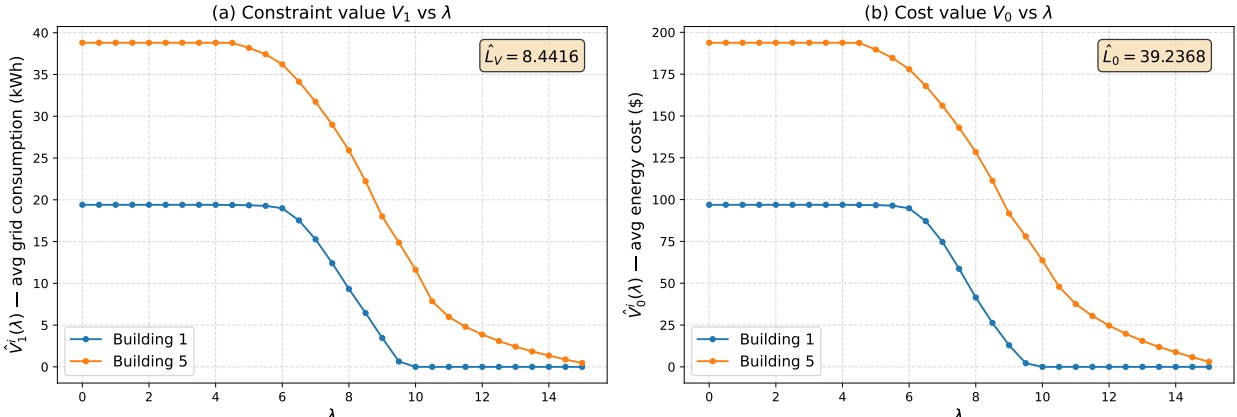

Figure B.1: Empirical sensitivity of value functions to the dual variable $\lambda$ ($\nu = -10$ fixed, 10 rollouts per point, horizon $= 80$). **(a)** Constraint value $V_1^i(\hat{\pi}^i(\lambda))$ (grid consumption) decreases monotonically with $\lambda$, with maximum slope $\hat{L}_V = 8.44\,\text{kWh}/\lambda$. **(b)** Cost value $V_0^i(\hat{\pi}^i(\lambda))$ shows similar behavior with $\hat{L}_0 = 39.24\,\$/\lambda$. Both curves are smooth, confirming Assumption 5.2.

*Remark* B.2 (Interpretation and Practical Implications). The bound in Proposition 5.4 decomposes the total primal error into three interpretable components: (i) *consensus error* ($L_V \sqrt{N}\,\delta$), the cost of agents disagreeing on the dual variable, controlled by the communication topology and the number of consensus rounds $\mathscr{L}$; (ii) *dual convergence error* ($L_V N |\bar{\lambda} - \lambda^\star|$), the distance of the average multiplier from the centralized optimum, which converges to zero under standard diminishing step-size conditions; and (iii) *function approximation error* ($N \varepsilon_{\text{approx}}$), the price of using neural network policies instead of exact optimizers.

In the idealized limit where $\bar{\lambda} = \lambda^\star$ and $\varepsilon_{\text{approx}} = 0$, the constraint violation reduces to $L_V \sqrt{N}\,\delta$, which decays exponentially in $\mathscr{L}$. For well-connected graphs ($\rho \ll 1$), even $\mathscr{L} = 1$ consensus round per iteration suffices to make this term negligible—consistent with the empirical finding in Section 7 that a single consensus step per time step achieves feasible solutions.

The $\sqrt{N}$ factor is a consequence of the Cauchy–Schwarz step and is tight in the worst case (all agents deviating coherently from the mean). In practice, the deviations tend to be incoherent across agents, and the empirical scaling is often milder.

*Remark* B.3 (Role of Function Approximation). In the tabular (exact optimization) setting with finitely many deterministic policies, the optimal policy $\pi_\star^i(\lambda)$ is piecewise constant in $\lambda$, making $V_1^i(\pi_\star^i(\lambda))$ piecewise constant with potential jump discontinuities at finitely many breakpoints. Assumption 5.2 would not hold globally in this case. However, with state-augmented neural network policies—where $\lambda$ is a continuous input to a smooth function approximator—the induced map $\lambda \mapsto \pi_\theta^i(\cdot \mid \cdot, \lambda)$ is inherently smooth, and the Lipschitz condition is natural. This is a setting where function approximation is not merely a computational convenience but a structural regularizer that enables the sensitivity analysis.

