# OpenReview forum: "Scalable Constrained Multi-Agent Reinforcement Learning via State Augmentation and Consensus for Separable Dynamics"
_TMLR — Accepted by TMLR_

### Review · Reviewer_LRZG · 2026-01-26

**Summary Of Contributions:**

The paper proposes a new distributed constrained multi-agent reinforcement learning (CMARL) method, provides theoretical analysis and proofs, and validates its effectiveness through experiments on demand response in smart grids. Moreover, it is commendable that the authors conduct a detailed comparative discussion with several related works, which is informative and worth referencing. However, the summary of the contributions and the discussion of related work remain insufficient, and there are also some typographical errors and formatting issues.

**Audience:**

Yes

**Audience Explanation:**

Although I am still uncertain about the significance of conducting research under the strong assumptions made in this paper, the theoretical analysis and proofs presented are novel and may attract interest from some researchers.

**Claims And Evidence:**

No

**Claims Explanation:**

1. The key contributions listed in the Introduction should be re-summarized. The current presentation appears inaccurate and incomplete in reflecting the actual content of the paper.
2. The paper explicitly assumes that state transitions and reward functions are mutually independent across agents, and that each constraint function can be replaced by either a local or global average. Under these assumptions, it is unclear what essential distinction this fully decentralized constrained reinforcement learning framework offers compared to simpler alternatives. Specifically, it raises the question of whether the proposed method unnecessarily overcomplicates a relatively straightforward problem.

**Requested Changes:**

1. There are instances of imprecise or inconsistent wording. For example, in the Introduction, the authors state: “Generally, MARL addresses a sequential problem …”, but then immediately follow with: “However, MARL problems …”, which creates a logical contradiction. Additionally, in the Related Work section, the sentence “average constraint satisfaction such as in our work (Liang et al., 2018; Paternain et al., 2022)” appears to contain a significant typographical error—Liang et al. (2018) and Paternain et al. (2022) are external references and should not be referred to as “our work.”
2. The Related Work section lacks recent references on Cooperative MARL and Networked MARL. For instance, the latest citation in the discussion of Cooperative MARL dates back to 2018, which is insufficient. The coverage of this important area is also inadequate.
3. Table 2 should be presented in a more visually appealing and reader-friendly format.
4. The reference formatting is inconsistent and does not adhere to a unified style. The authors are advised to carefully review and standardize the bibliography format throughout the manuscript.

---

> ### Author Response · Authors · 2026-02-12
> **Response to Reviewer LRZG**
>
> # Response to Reviewer LRZG
>
> **Reviewer:** The key contributions listed in the Introduction should be re-summarized. The current presentation appears inaccurate and incomplete in reflecting the actual content of the paper.
>
> **Response:** We have revised the contributions in Section 1:
> * A distributed consensus mechanism over Lagrange multipliers integrated with state-augmented constrained RL, enabling global constraint coordination through local communication, with provable bounded consensus error;
> * Linear scalability demonstrated with up to 1000 agents (vs. $\sim$10--20 for CTDE methods);
> * Empirical evidence that consensus is necessary for feasibility---identical policies produce feasible vs. infeasible outcomes depending solely on whether consensus is enabled.
>
> ---
>
> **Reviewer:** The paper explicitly assumes that state transitions and reward functions are mutually independent across agents, and that each constraint function can be replaced by either a local or global average. Under these assumptions, it is unclear what essential distinction this fully decentralized constrained reinforcement learning framework offers compared to simpler alternatives. Specifically, it raises the question of whether the proposed method unnecessarily overcomplicates a relatively straightforward problem.
>
> **Response:** Section 7.2 provides direct empirical evidence. The constraint $\sum_{i=1}^{N} V_1^i(\pi^i) \leq c$ creates coupling in the *feasible policy space*: one agent's policy affects whether others can satisfy the global constraint. This cannot be resolved independently.
>
> Our experiments demonstrate:
> * **Same policies, different outcomes**: Identical trained policies produce feasible or infeasible behavior depending solely on consensus (Figures 3--4).
> * **Multiplier saturation**: Without consensus, $\lambda^i$ diverge to upper bounds (Figure 4b), indicating local updates cannot find a feasible point.
> * **Degenerate solutions**: The no-consensus variant satisfies the grid constraint by indefinitely postponing demand, failing $V_2^i = 0$.
>
> Satisfying $\sum_i V_1^i \leq c$ requires coordinating constraint usage; without communication, agents must guess others' consumption. Consensus resolves this distributed fixed-point problem. We revised the Abstract and Introduction to emphasize this finding.
>
> ---
>
> **Reviewer:** There are instances of imprecise or inconsistent wording. For example, in the Introduction, the authors state: "Generally, MARL addresses a sequential problem...", but then immediately follow with: "However, MARL problems...", which creates a logical contradiction. Additionally, in the Related Work section, the sentence "average constraint satisfaction such as in our work (Liang et al., 2018; Paternain et al., 2022)" appears to contain a significant typographical error, Liang et al. (2018) and Paternain et al. (2022) are external references and should not be referred to as "our work."
>
> **Response:** The sentence has been corrected to: "average constraint satisfaction as studied in previous work (Liang et al., 2018; Paternain et al., 2022)." We also revised the Introduction to eliminate the logical inconsistency and reviewed the manuscript for similar issues.
>
> ---
>
> **Reviewer:** The Related Work section lacks recent references on Cooperative MARL and Networked MARL. For instance, the latest citation in the discussion of Cooperative MARL dates back to 2018, which is insufficient. The coverage of this important area is also inadequate.
>
> **Response:** We expanded Related Work to include recent advances through 2025: surveys by Kungurtsev et al. (2025), Low and Zhou (2025), and Hady et al. (2025); the textbook by Albrecht et al. (2024); consensus-based MARL by Oh et al. (2025) and Li et al. (2025); and power systems control by Cui et al. (2023) and Feng et al. (2023).
>
> We now explicitly position against recent consensus-based MARL: Oh et al. and Li et al. use consensus for *reward aggregation* in unconstrained settings, while ours operates on *dual variables* for constraint satisfaction---a fundamentally different problem where independent learning fails.
>
> ---
>
> **Reviewer:** Table 2 should be presented in a more visually appealing and reader-friendly format.
>
> **Response:** We reformatted Table 2 with category groupings, midrules, and a footnote distinguishing scalable unconstrained methods from constrained coupled-dynamics methods.
>
> ---
>
> **Reviewer:** The reference formatting is inconsistent and does not adhere to a unified style. The authors are advised to carefully review and standardize the bibliography format throughout the manuscript.
>
> **Response:** We have reviewed and standardized all bibliography entries for consistent formatting, aviding to the style guides of TMLR.

---

### Review · Reviewer_eNNH · 2026-01-27

**Summary Of Contributions:**

1. Proposed a scalable framework for constraiend multi-agent system by reducing multi-agent learning to single-agent constrained RL with state augmentation
2. Introduced a distributed dual-consensus mechanism that coordinates agents through local communication without centralized critics
3. Demonstrates larg-scale execution

**Additional Comments:**

Update after rebuttal:
Thanks for the great rebuttal. The revision is competent and addresses most of my concerns honestly. However, the limitation remains, limited scope and generalizability. The reviewer will suggest a boardline accept.

**Audience:**

No

**Audience Explanation:**

General applicability beyond the specific energy management is hard. The scope is too limited.
1. It is not clear why the constraints should be shared evenly across agents. Different agents should have flexibility. In another word, separable dynamics do not justify separable or evenly split constraints.
2. The assumption is hard to satisfy. For example car congestion example, the move of one agent indeed change the congestion time of other cars. The seperation in real applications will have a very small niche or needs stronger assumptions.
3. Policy learning is entirely single-agent. Dual consensus changes the system dynamics and remains active at test time. This might influence the performance of trained policy.

**Claims And Evidence:**

No

**Claims Explanation:**

The experimental evidence is accurate but incomplete. It is feasible and scalable, but does not convincingly validate several key assumptions.
Claims supported: Dual consensus is needed. Can scale.
Not supported:
1. Effectiveness of state augmentation. What if it is dropped?
2. With consensus, negative unmet demand is also undesirable and not discussed.
3. Reliablity of learned safety value function. It is assumed in Theorem 5.1, but how accurate is the value function estimated.

**Requested Changes:**

Check the beyond concerns.
1. Presentation. Figure 5 is not well-motivated, seems not necessary.
2. Clarify problem scope and positioning. For consensus, there is also a lot paper do that, such as:
Structured neural-pi control with end-to-end stability and output tracking guarantees, this work use a neural pi to solve economic-dispatch, or Bridging transient and steady-state performance in voltage control: A reinforcement learning approach with safe gradient flow use optimization, also make things local, they are both in power system control domain. Not exactly same but share a global cost point of view.
3. Justify or relax even constraint decomposition.
4. Assess reliability of the safety value function V1.
5. Comment on over-serving and variance issues.

---

> ### Author Response · Authors · 2026-02-12
> **Response to Reviewer eNNH**
>
> **Reviewer:** Effectiveness of state augmentation. What if it is dropped?
>
> **Response:** Section 7.2 addresses this via an ablation over 414 fixed-$(\lambda, \nu)$ configurations (i.e., policies *without* augmentation). **Only 8 ($<2\%$) produced feasible policies**, confirming Calvo-Fullana et al. (2023, IEEE TAC). Fixed-$(\lambda,\nu)$ policies cannot adapt during execution, whereas $\pi(s,\lambda)$ responds to consensus-driven updates. We revised Section 7.2's title and framing accordingly.
>
> ---
>
> **Reviewer:** With consensus, negative unmet demand is also undesirable and not discussed.
>
> **Response:** The negative unmet demand in Figure 6a is not "over-serving" but **proactive load shifting**: agents charge batteries during solar peaks and pre-serve demand using stored energy. The **stable negative value** ($\approx -350$ kWh for $c=0.30$) indicates a healthy equilibrium. The key metric is **stability**, not sign: stable values indicate feasibility; unbounded growth indicates failure. We added Remark 7.1 and revised Figure 6a's caption.
>
> ---
>
> **Reviewer:** Reliability of learned safety value function. It is assumed in Theorem 5.1, but how accurate is the value function estimated.
>
> **Response:** Theorem 5.1 requires *bounded* (not perfect) estimates: $\|V_1^i - \hat{V}_1\| \leq \sigma$ holds for any bounded reward by standard arguments. Consensus error remains bounded *even with imperfect estimates*. Monte Carlo returns provide unbiased $V_1^i$ estimates, and consensus smooths variance through averaging. We added text clarifying these robustness guarantees.
>
> ---
>
> **Reviewer:** It is not clear why the constraints should be shared evenly across agents. Different agents should have flexibility.
>
> **Response:** The $c/N$ in Equation (7) arises from algebraic manipulation of the Lagrangian, not a design choice. It **does not** enforce even splitting: the global constraint $\sum_i V_1^i \leq c$ is maintained, and consensus enables heterogeneous usage---an agent consuming more than $c/N$ is compensated by others consuming less. This flexibility is inherent in consensus-based coordination.
>
> ---
>
> **Reviewer:** The assumption is hard to satisfy. For example car congestion, the move of one agent changes congestion for others.
>
> **Response:** We agree and revised Remark 3.4 accordingly, referencing methods for coupled settings (Lu et al., 2021; Zhang et al., 2024). *When separable structure holds*---smart grids, distributed computing, fleet management, warehouse robotics---scalability to 1000 agents is achievable versus $\sim$10--20 for coupled-dynamics methods, a $100\times$ improvement for the applicable problem class.
>
> ---
>
> **Reviewer:** Policy learning is entirely single-agent. Dual consensus changes the system dynamics and remains active at test time.
>
> **Response:** $\pi^i(s, \lambda)$ *explicitly takes $\lambda$ as input* (state augmentation). Consensus does not change MDP dynamics; it governs $\lambda$'s evolution within the augmented state. Training with uniform $\lambda$ sampling ensures the policy responds correctly to consensus-driven $\lambda$ evolution at test time. Without augmentation, policies fail when $\lambda$ changes: $<2\%$ of fixed-$\lambda$ policies are feasible in our ablation.
>
> ---
>
> **Reviewer:** Figure 5 is not well-motivated, seems not necessary.
>
> **Response:** Figure 5 provides central ablation evidence (8/414 fixed-multiplier configurations feasible). We retained it and improved its motivation in revised Section 7.2.
>
> ---
>
> **Reviewer:** Clarify problem scope and positioning. References: Structured neural-pi control (Cui et al., 2023) and Bridging transient and steady-state performance in voltage control (Feng et al., 2023).
>
> **Response:** These works develop neural controllers with stability guarantees for voltage control. Key differences: (1) they address output tracking via structured neural PI control; we address constraint satisfaction via RL; (2) their constraints use strictly convex neural networks; ours use Lagrangian duality with consensus; (3) they require dynamics knowledge; we are model-free. The approaches are complementary. We added these references to Related Work.
>
> ---
>
> **Reviewer:** Justify or relax even constraint decomposition.
>
> **Response:** See above: $c/N$ arises from Lagrangian algebra, not equal-share enforcement. Consensus inherently allows heterogeneous constraint usage.
>
> ---
>
> **Reviewer:** Assess reliability of the safety value function V1.
>
> **Response:** Addressed above regarding Theorem 5.1's robustness guarantees.
>
> ---
>
> **Reviewer:** Comment on over-serving and variance issues.
>
> **Response:** Addressed above: negative unmet demand is intentional load shifting, and Theorem 5.1 bounds consensus error under estimation variance.

---

### Review · Reviewer_RLhF · 2026-03-09

**Summary Of Contributions:**

This paper studies constrained multi-agent reinforcement learning in a restricted but practically relevant regime: agents use independent local policies, have separable dynamics, and the global objective decomposes additively, while coupling occurs through a global average constraint. The proposed method trains state-augmented local policies offline, conditioning each policy on a dual variable, and then coordinates execution by updating local copies of the multiplier through neighbor-to-neighbor consensus. The paper also provides a bounded-consensus-error result for the distributed multiplier dynamics and evaluates the approach on smart-grid demand response, where the experiments are intended to show that consensus avoids a degenerate solution that satisfies the grid cap only by postponing demand. The execution study is reported up to 1000 agents.

**Audience:**

Yes

**Audience Explanation:**

Yes. I do think a meaningful subset of TMLR readers would care about this paper. Scalable constrained RL and multi-agent coordination under global resource limits are important topics, and the paper targets a practically relevant class of infrastructure-management problems such as smart-grid demand response. Even though the setting is restricted to separable dynamics, that restriction is made explicit in the paper, and the combination of state-augmented constrained RL with distributed dual consensus is a useful idea for this situation.

**Broader Impact Concerns:**

I do not have major broader impact concerns.

**Claims And Evidence:**

No

**Claims Explanation:**

Not yet.

The theory supports a narrower conclusion than some of the paper's strongest claims suggest. Theorem 5.1 gives a bounded consensus-error result under the extra assumption that local constraint values stay within $\sigma$ of their mean, but the text then interprets this as near-centralized optimality and satisfaction of the global constraints. I did not find a direct end-to-end theorem giving a primal feasibility gap or optimality gap for the practical algorithm, and the experiments add a local demand-fulfillment constraint $V_2^i\left(\pi_i\right)=0$ with multiplier $\nu_i$ that is not clearly covered by the main theory. In the appendix, the step from the condition-number-based bound in Theorem A. 5 to the simplified $\left\|P^L\right\|=\rho^L$ style argument in Theorem A. 7 would also benefit from a clearer explanation.

The empirical evidence is promising, but not fully convincing enough for the strongest empirical claims. The paper itself notes that broader evaluation would strengthen the claims. The baseline comparison selects a fixed penalty pair from an exhaustive search, then reports the rollout trajectory closest to the constraint threshold among 10 roll-outs for MAPPO, MADDPG, MASAC, and ISAC. That is a relatively weak constrained RL protocol, especially compared with what an adaptive constrained baseline or centralized dual-variable coordinator could show. Likewise, the scaling experiment directly validates execution-time scaling to 1000 agents, but not end-to-end training scalability, even though the abstract and conclusion claim linear scaling in both training and execution.

That said, the paper does provide good evidence for narrower claims: the with/without-consensus ablation clearly shows degenerate demand postponement without coordination, and the state-augmentation ablation is a genuine strength of the submission.

**Requested Changes:**

1. [Critical] Please reconcile the constraint direction and dual-update sign across Sections $3-6$, Algorithm 1, and the smart-grid use case. Right now the general formulation uses $\sum_i V_1^i\left(\pi_i\right) \geq c$ and Eq. (5), while Algorithm 1 and Section 6 behave like a $\leq c$ constraint on grid usage.

2. [Critical] Please either prove or tone down the strongest claims. The current theory gives bounded consensus error under additional assumptions, but the text makes broader claims about near-optimality, satisfaction of global constraints, and independent learning "provably" failing. A direct feasibility/optimality result, or more modest wording, would materially improve the paper.

3. [Critical] Please clarify how the practical algorithm maps to the theory. In particular, please define how $V_{1, k}^i$ in Algorithm 1 is instantiated during execution, explain the choice of $\alpha, \epsilon, \eta, L$.

4. [Critical] Please either support the training-scaling claim with measurements or narrow the claim. Figure 7 is explicitly about the execution phase, so the manuscript currently provides direct evidence only for execution-time scaling. Training wall-clock, memory, or at least a more careful complexity discussion would be needed.

5. [Would strengthen] The current baseline study mainly shows the brittleness of fixed penalties. Since the related work section already discusses constrained MARL methods such as Safe Dec-PG, MACPO, and Scal-MAPPO-L, I would like to see at least one fair constrained MARL baseline on a small setting where such a comparison is computationally feasible. Otherwise, please provide an explanation for why it isn't comparable.

---

> ### Author Response · Authors · 2026-03-17
> **Response to Reviewer RLhF**
>
> We thank the reviewer for the constructive feedback.
>
> **Reviewer** Please reconcile the constraint direction and dual-update sign across Sections 3-6, Algorithm 1, and the smart-grid use case. Right now the general formulation uses $\sum_i V_1^i\left(\pi_i\right) \geq c$ and Eq. (5), while Algorithm 1 and Section 6 behave like a $\leq c$ constraint on grid usage.
>
> **Response** Unified to $\leq$ throughout Sections 3–4 (Eqs. 1b, 3b, 4, 5, 6, 7, 8, 9) and narrative. The Lagrangian is now $\mathcal{L} = \sum V_0^i + \lambda(c - \sum V_1^i)$ and the weighted reward $r_\lambda = r_0 - \lambda r_1$, consistent with Algorithm 1 and the smart-grid use case. Algorithm 1 and appendix proofs were already correct and unchanged.
>
> ---
>
> **Reviewer:** Please either prove or tone down the strongest claims. The current theory gives bounded consensus error under additional assumptions, but the text makes broader claims about near-optimality, satisfaction of global constraints, and independent learning "provably" failing. A direct feasibility/optimality result, or more modest wording, would materially improve the paper.
>
> **Response:** Two actions. First, we added Proposition 5.4 + Corollary 5.5: under a Lipschitz assumption on $\lambda \mapsto V_1^i(\pi^\star(\lambda))$ (Assumption 5.2) and an approximate optimality assumption (Assumption 5.3), the consensus error $\delta$ from Theorem 5.1 implies constraint violation $\leq c + L_V\sqrt{N}\,\delta + N\,\varepsilon_{\mathrm{approx}}$. The proof (Appendix B) decomposes the error via triangle inequality into consensus, dual convergence, and function approximation terms, using Cauchy–Schwarz to bridge $\ell_2$ consensus error to $\ell_1$ agent deviations. The Lipschitz constant is justified theoretically (Danskin's theorem, standard MDP sensitivity results) and measured empirically: $\hat{L}_V = 8.44$ kWh/$\lambda$ (new Figure B.1), yielding a non-vacuous bound for our 7-agent setting. Second, we softened language: "provably fails" $\to$ empirical; "near-optimal" $\to$ "bounded constraint violation"; updated abstract, contributions, and related work.
>
> ---
>
> **Reviewer:** Please clarify how the practical algorithm maps to the theory. In particular, please define how $V_{1, k}^i$ in Algorithm 1 is instantiated during execution, explain the choice of $\alpha, \epsilon, \eta, \mathscr{L}$.
>
> **Response:** Added Remark 5.7: $V_{1,k}^i = r_1^i(s_k^i, a_k^i)$ is the instantaneous constraint reward, serving as a single-sample stochastic estimate of $V_1^i(\pi_\star^i(\lambda_k^i))$. The local multiplier $\nu^i$ enforces per-agent demand fulfillment (Eq. 12), does not couple agents, and does not affect the consensus analysis in Theorem 5.1 or Proposition 5.4. Additionally, we now report all step sizes ($\alpha = \epsilon = \eta = 0.01$, $\mathscr{L} = 1$) with justification: the consensus bound shows asymptotic error scales as $\alpha\sigma/(1-\rho)$, linking hyperparameter choice to theoretical guarantees. We also clarified in the appendix proof why the condition number $C = \kappa(V)$ from Theorem A.5 does not appear in Theorem A.7: the error vector lies in the subspace orthogonal to $\mathbf{1}$ by construction, so $\|P^{\mathscr{L}}\| = \rho^{\mathscr{L}}$ directly.
>
> ---
>
> **Reviewer** Please either support the training-scaling claim with measurements or narrow the claim. Figure 7 is explicitly about the execution phase, so the manuscript currently provides direct evidence only for execution-time scaling. Training wall-clock, memory, or at least a more careful complexity discussion would be needed.
>
> **Response** Training cost is $O(1)$ in $N$: one policy per agent type, same two policies serve 7–1000 agents. Measured: 26.7 min total (13.2 + 13.5), $\sim$5 GB peak memory, $10^6$ PPO steps each. CTDE methods require retraining whenever $N$ changes. New paragraph in Section 7.4.
>
> ---
>
> **Reviewer** The current baseline study mainly shows the brittleness of fixed penalties. Since the related work section already discusses constrained MARL methods such as Safe Dec-PG, MACPO, and Scal-MAPPO-L, I would like to see at least one fair constrained MARL baseline on a small setting where such a comparison is computationally feasible. Otherwise, please provide an explanation for why it isn't comparable.
>
> **Response** These methods address different problem settings (coupled dynamics, per-step or cumulative safety constraints) and are not directly applicable to our separable-dynamics, average-constraint formulation. Rather than adapting an external method to our setting, we compare against a centralized oracle: a single global $\lambda$ updated with perfect knowledge of the aggregate constraint signal. This is the strongest possible baseline under our formulation, as it represents the performance achievable with unlimited communication. Over 10 paired seeds (3000 steps): very similar grid consumption (44.89 kWh), cost gap $-0.01\% \pm 0.13\%$, constraint satisfied 100% for both. New Table 3 and Figure 8.

---

### Author Response · Authors · 2026-02-12
**General Response**

# General Comment to Reviewers

We thank both reviewers for their constructive feedback. Their comments have helped us strengthen the paper's presentation and clarify our contributions. Below we address each concern in detail, describing the corresponding revisions made to the manuscript. Every point raised by the reviewers has been taken into account and addressed in the revised manuscrip, including presentation improvements, expanded references, and clarified theoretical and empirical claims.

We apologize for the concise and direct tone of our individual responses; this is due to the limited number of characters available in the response fields. To supplement those answers, we offer the following clarifications:

**On state augmentation and policy training (Reviewer eNNH):** During training, the multiplier $\lambda$ is sampled uniformly so that the policy learns to behave optimally for *any* multiplier value. This is what enables the policy to respond correctly to the consensus-driven evolution of $\lambda$ at test time, and is the core reason why state augmentation succeeds where fixed-multiplier policies fail (only 8 of 414 configurations, $<2\%$, produced feasible behavior in our ablation).

**On why consensus is necessary despite separable dynamics (Reviewer LRZG):** Although agents have independent transitions and rewards, the global constraint $\sum_i V_1^i \leq c$ couples the *feasible policy space*: each agent must implicitly account for how much of the shared constraint budget others consume. Without communication, agents have no mechanism to coordinate this usage. Consensus on the dual variables provides exactly this coordination, resolving what is essentially a distributed fixed-point problem.

**On the suggested references (both reviewers):** We have incorporated all suggested references (Cui et al., 2023; Feng et al., 2023) as well as recent surveys and consensus-based MARL works (Oh et al., 2025; Li et al., 2025; Kungurtsev et al., 2025; among others) into the revised Related Work section, with explicit discussion of how our approach differs from each.

---

### Author Response · Authors · 2026-05-29

# Response to the Action Editor

We thank the Action Editor for the careful recommendation. We have taken the five points raised seriously and addressed each in the camera-ready manuscript. The corresponding changes are summarized below.

**1. Sharpened theoretical claims.** The Conclusion has been rewritten to state explicitly that the empirical near-equivalence with the centralized oracle is an observation in our smart-grid testbed, not a general optimality claim. The new Appendix B (*Sensitivity Analysis: From Consensus Error to Feasibility*) and Proposition 5.4 make the connection between the consensus error bound and the feasibility margin precise, and the absence of a full primal optimality guarantee is stated explicitly.

**2. Local demand-fulfillment constraint and its multiplier.** The Conclusion now states that Theorem 5.1 and Proposition 5.4 concern only the global multiplier $\lambda$. The local-constraint multiplier $\nu$ is described as updated by an analogous primal-dual scheme that is well-behaved empirically but is not covered by the formal guarantees, and we treat it as an empirical extension of the global-constraint theory rather than a setting in which the analysis applies.

**3. Stronger baselines.** Following the second option in the recommendation, Section 7.3 (*Centralized Oracle Comparison*) has been added, with a paired 10-seed evaluation against an adaptive centralized dual baseline that uses the true aggregate constraint satisfaction; results appear in Table 1 and the corresponding four-panel figure. Section 7.2 has been extended with the IPPO penalty-grid ablation and the BenchMARL comparison (MAPPO, MADDPG, MASAC, ISAC).

**4. Scalability decomposition.** Section 7.4 now distinguishes three quantities explicitly: training complexity is $O(1)$ in the number of agents, since a single policy per agent type serves all configurations from 7 to 1{,}000 agents; execution complexity is linear in the number of agents; and empirical wall-clock and memory measurements are reported (26.7 minutes total training on a 16-core CPU with approximately 5 GB peak memory). Scaling is demonstrated up to 1{,}000 agents.

**5. Limitations and applicability.** The Conclusion has been rewritten to identify the class of problems to which the framework applies and the regime in which it does not. We name fitting application classes (smart-grid demand response, distributed electric-vehicle charging, traffic-signal coordination at lightly-coupled intersections, water- and heating-network management) along with the structural reason they fit (per-node controllable resources, linearly-aggregated global constraint), and non-fitting cases (multi-robot manipulation, cooperative navigation in a shared workspace, pursuit-evasion in a joint state space) tied to violations of Assumption 3.2. We further acknowledge that the empirical evidence presented comes predominantly from one application domain and that broader validation across the application classes identified above is a priority for follow-up work.

Sincerely,
The Authors

---

### Decision · Action_Editor_wBHz · 2026-05-10

**Recommendation:** Accept with minor revision

**Additional Comments:**

For the revision, the authors should focus on sharpening the scope, claims, and evidence. First, the main theoretical result should be presented as precisely as possible, with a clear statement of what is and is not guaranteed for the practical algorithm. If a full primal feasibility or optimality result is not available, the text should avoid suggesting near-centralized optimality or general constraint satisfaction beyond the established bounds. Second, the authors should further clarify how the local demand-fulfillment constraint and its multiplier relate to the main theory. Third, the empirical section should more directly support the main claims: either include stronger constrained-MARL or centralized/adaptive-dual baselines on smaller feasible instances, or clearly justify why such comparisons are inappropriate. Fourth, the scalability claim should distinguish training complexity, execution complexity, and empirical wall-clock/memory measurements. Finally, the authors should further discuss the limitations of the separable-dynamics assumption and identify the class of applications for which the proposed method is expected to be appropriate.

While the recommendation is "minor revision", I would like to emphasize that the authors have to address the above issues clearly in the revision.

**Audience:**

Yes

**Audience Explanation:**

I believe that at least some individuals in TMLR’s audience would be interested in the findings of this paper. Scalable constrained MARL, decentralized coordination, and safe resource allocation are topics of clear relevance to parts of the TMLR community. Even though the paper focuses on a restricted setting with separable dynamics and global average constraints, this setting is meaningful for applications such as smart-grid demand response and related infrastructure-management problems.

**Claims And Evidence:**

Yes

**Claims Explanation:**

**Summary**: The paper studies constrained multi-agent reinforcement learning for systems with separable agent dynamics and globally coupled resource constraints. The proposed approach trains local policies augmented with a dual variable and uses neighbor-to-neighbor consensus over local Lagrange multipliers at execution time to coordinate constraint satisfaction without centralized critics. The paper provides theoretical analysis of bounded consensus error and relates this error to constraint violation under additional assumptions. Empirically, the method is evaluated on a smart-grid demand-response setting, where the authors argue that consensus avoids degenerate behavior observed in independent learning and enables scaling to large numbers of agents.


The reviewers agree that the paper has a clear technical direction and that the rebuttal/revision addressed several concrete concerns, including the constraint-sign inconsistency, clarification of the role of state augmentation, improved positioning with respect to related work, and additional discussion of scaling and baselines. Distributed dual consensus can be useful in the considered separable-dynamics, globally constrained setting, and the smart-grid experiments provide encouraging support for this claim. At the same time, the paper’s broader claims about scalability, feasibility guarantees, and applicability remain stronger than what is fully established. The theoretical guarantees rely on additional assumptions, and the connection between the consensus-error analysis, the practical algorithm, and the full constrained MARL objective should be made more explicit. The empirical study is promising but still limited in scope, with most evidence coming from one application domain, and the comparison to constrained MARL or centralized/adaptive dual baselines remains an important point to strengthen.

**Resubmission Of Major Revision:**

The authors may consider submitting a major revision at a later time.